# Increased urea nitrogen salvaging by a remodeled gut microbiota helps nonhibernating pikas maintain protein homeostasis during winter

Fuyu Shi[1], Desheng Zou[1], Liangzhi Zhang[2], Na Guo[1], Jiangkun Yu[3], Abraham Allan Degen[4], Xianjiang Tang[2], Shien Ren[2], Yuning Ru[1], Shuai Zheng[1], Yanming Zhang[2]*, Dehua Wang[1]*

1 School of Life Sciences, State Key Laboratory of Microbial Technology, Shandong University, Qingdao, Shandong, China, 2 Key Laboratory of Adaptation and Evolution of Plateau Biota, Northwest Institute of Plateau Biology, Chinese Academy of Sciences, Xining, Qinghai, China, 3 State Key Laboratory for Conservation and Utilization of Bio-Resources in Yunnan, School of Life Sciences, Yunnan University, Kunming, Yunnan, China, 4 Blaustein Institutes for Desert Research, Ben-Gurion University of the Negev, Beer Sheva, Israel

* zhangym@nwipb.cas.cn (YZ); dehuawang@sdu.edu.cn (DW)

## Abstract

Nitrogen balance is a major challenge for herbivores when consuming a low-nitrogen diet. Gut microbiota-mediated urea nitrogen recycling facilitates protein homeostasis during times of nitrogen deficiency, yet its relevance to wild nonhibernating small mammals remains unclear. Here, through a combination of isotope tracing, metagenomics, targeted short-chain fatty acid analysis, and fecal microbiota transplantation, we investigated the effects of protein restriction in winter on urea nitrogen recycling in plateau pikas (*Ochotona curzoniae*) of the Qinghai–Tibetan Plateau. Hepatic urea-cycle metabolism was downregulated during winter protein restriction, accompanied by increases in beneficial bacteria with ureolytic capacity (such as the genus *Alistipes*), gut urease activity, and urea transporters, and acetate production, with a consequent increase in nitrogen reincorporation into the pika's protein pool. Critically, supplementing a low-protein diet with yak fecal microbiota enhanced the ureolytic capacity by increasing *Alistipes* abundance, revealing a critical mechanism whereby interspecies horizontal microbial transfer between sympatric species enhances host protein homeostasis. Our results reveal a functional role for the gut microbiota in urea nitrogen recycling to maintain protein balance in winter-active herbivorous small mammals and contribute to our understanding of species coexistence and mammalian adaptation to high-altitude environments. Our findings establish that microbiota-driven urea nitrogen recycling is a key adaptive strategy for protein homeostasis in winter-active herbivores. This work provides new insights into the mechanisms of mammalian adaptation to high-altitude environments and the dynamics of interspecies coexistence.

which permits unrestricted use, distribution, and reproduction in any medium, provided the original author and source are credited.

**Data availability statement:** The raw metagenomic sequencing data are available in the National Center of Biotechnology Information (NCBI) Sequence Read Archive (SRA) under accession number PRJNA1190225 (https://dataview.ncbi.nlm.nih.gov/object/PRJNA1190225?reviewer=4q92h87i30nqhcmmk7h-6heo64s). The MAGs and their functional annotations generated in this study have been deposited in the Figshare repository under DOI: https://doi.org/10.6084/m9.figshare.29646257. All custom R scripts used for data analysis and visualization are publicly available on GitHub (https://github.com/FuyuShi1122/urea-nitrogen-recycling) and have been archived on Zenodo (DOI: https://doi.org/10.5281/zenodo.17120073). All other relevant data are included within the article and its Supporting information files.

**Funding:** This work was supported by the National Natural Science Foundation of China (http://www.nsfc.gov.cn/) to FYS (No.32301301) and DHW (No.32330012), the Double-First Class Initiative of the School Life Sciences, Shandong University (http://www.sdu.edu.cn/) to DHW, and the Shandong Province Natural Science Foundation (http://kjt.shandong.gov.cn/) to FYS (ZR2024QC355). The funders had no role in study design, data collection and analysis, decision to publish, or preparation of the manuscript.

**Competing interests:** The authors have declared that no competing interests exist.

**Abbreviations:** Abx, antibiotic cocktail; *Arg1*, Arginase 1; *Asl*, Argininosuccinate lyase; *Ass1*, Argininosuccinate synthase 1; $CO_2$, carbon dioxide; CP, carbamyl phosphate; CP, crude protein; CPS1, carbamoyl phosphate synthetase 1; *Eaat3*, excitatory amino acid transporter 3; FDR, false-discovery rate; FMT, fecal microbiota transplantation; HP, high protein; IRMS, isotope ratio mass spectrometry; KEGG, Kyoto Encyclopedia of Genes and Genomes; *Lat1*, L-type amino acid transporter 1; LP, low protein; MAG, metagenome-assembled genome; MP, medium protein; $NH_3$, ammonia; NMDS, nonmetric multidimensional scaling; OTC, ornithine transcarbamylase; QTP, Qinghai–Tibetan Plateau; SCFA, short-chain fatty acid; UT-B, urea transporter protein.

## Introduction

Many winter-active mammals inhabiting cold, high-altitude regions require large amounts of nutrients and energy. However, the increased energy needs of mammals in winter are concomitant with a reduced nutritive value of available forage. This situation creates a challenge for small mammals to meet their energy and nitrogen requirements, in particular those that do not migrate or hibernate, such as the plateau pika [1–3]. When forced to select between protein and energy intake during a shortage of nutrient supply, wild animals prioritize protein over energy, a behavior known as the protein leverage effect [4,5]. Chronic protein deficiency usually accelerates muscle loss and decelerates animal growth [6,7]. Winter-active herbivores consume nitrogen-poor food that contains large amounts of structural carbohydrates during winter. Despite dietary nitrogen deficiency during winter, most mammals are able to maintain stable physiological functions, such as maintain muscle mass and enzyme activity [8,9]. Small hibernating mammals salvage urea nitrogen, such as thirteen-lined ground squirrel (*Ictidomys tridecemlineatus*), European hare (*Lepus europaeus*) and Muroid Rodents [10–12], however, it is uncertain whether nonhibernating small mammals do so as well.

Urea nitrogen salvaging functions as a key adaptive strategy in wild animals, countering resource limitations and environmental stressors through nitrogen conservation, energy optimization, and homeostatic regulation [13]. In this process, urea synthesized by the liver is transported into the gut lumen instead of being excreted. The transmembrane transport of urea from the bloodstream across the gut epithelium is mediated by urea transporter protein (UT-B). In the gut lumen, ureolytic microbes convert urea to ammonia ($NH_3$) and carbon dioxide ($CO_2$), and the $NH_3$ is incorporated into bacterial protein, which ultimately provides amino acids to the host. The abundance and efficiency of UT-B are regulated largely by microbial urease [14]. The breakdown of urea in the gut and the efficiency of $NH_3$ utilization depend mainly on the composition and function of ureolytic gut bacteria [15,16]. Therefore, the interaction between the ureolytic microbiota and urea transporters in the gut epithelium plays a critical role in regulating urea nitrogen recycling back into the host's protein pool.

The Qinghai–Tibetan Plateau (QTP) is known as "The Third Pole", with an average elevation of more than 4,000 m above sea level. It is the largest (2.5 million $km^2$) and highest plateau in the world and consists of different alpine grasslands, including desert steppes, alpine steppes, and alpine meadows [17]. The plateau is characterized by strong winds, intense solar radiation, low precipitation, and a short growing season of 120−150 days between May and September. Winter on the plateau is cold, with a mean annual air temperature of below 0°C and extremely low temperatures (below −30°C) from November to February [18]. Maximal summer biomass is ~2,900 kg dry matter per hectare (DM/ha), but winter biomass is only ~100 kg DM/ha [19]. In winter, the nutrient value of the forage is exceedingly poor, with reduced levels of crude protein (from 180 to 40 g/kg DM) but elevated neutral detergent fiber (from 563 to 682 g/kg DM, respectively) from summer to late winter [19].

The plateau pika (*Ochotona curzoniae*) is a small lagomorph distributed in alpine meadows on the QTP, ranging in elevation from 3,100 to 5,100 m above sea level. They survive the long, cold winter without hibernating or storing food [20], which differs from the American pika (*Ochotona princeps*) in the Rocky Mountains in North America that survives winter by caching forage underground [21]. Plateau pika must rely on poor-quality forage, and employ low energy requirements to maintain normal body temperature, stable body weight, and body fat content during winter [22–26]. Thus, this species is an ideal model for investigating winter-adaptive nutrition strategies. Evidence suggests that the European hare (*Lepus europaeus*) employs urea recycling from the liver as an adaptive mechanism to increase nitrogen utilization [11]. Plateau pikas are particularly sensitive to low dietary protein [27], a significant challenge during the harsh winters on the QTP. Evidence suggests that their native gut microbiota, particularly the diazotrophic community, may facilitates urea-nitrogen recycling through host-microbe metabolic cross-talk, enabling them to cope with seasonal protein deficiency [28,29]. However, pikas may have evolved an even more sophisticated strategy to survive extreme conditions. It is paradoxical that pika populations thrive in areas with high densities of their primary food competitor, the sympatric yak (*Bos poephagus*, a ruminant raised domestically on the QTP), especially during winter. This phenomenon is largely explained by a recently discovered survival strategy: interspecific coprophagy [26]. Strong evidence confirms that pikas consume yak feces, a relatively rare interspecific coprophagy that provides crucial nutrients when vegetation is scarce but also, more fundamentally, alters their gut microbiota [26,28]. While the general nutritional benefits are apparent, the specific functional consequences of this microbiota modulation for host nitrogen metabolism remain largely unexplored. Given that yak feces are rich in ureolytic bacteria essential for urea hydrolysis [19], we hypothesize that this interspecific coprophagy is a behavioral adaptation to boost the nitrogen-recycling capabilities of the pika's gut microbiota, enabling the plateau pikas to maintain protein homeostasis over the winter.

Therefore, we tested the central hypothesis that interspecific coprophagy is a key behavioral adaptation for pikas to overcome extreme winter nitrogen limitation by modulating their gut microbiome. This central hypothesis leads to three specific predictions: (1) the native gut microbiota of pika possesses an inherent capacity for urea-nitrogen recycling, and that this function is significantly upregulated in response to a low-protein (LP) diet; (2) pikas use interspecific coprophagy (experimentally simulated by supplementing an LP diet with yak fecal microbiota, i.e., the LPY diet) as a behavioral strategy to further augment this inherent capacity, by acquiring potent ureolytic microbes from yak feces; and (3) this microbial enhancement translates into a physiological benefit, namely the improved maintenance of protein homeostasis.

## Results

### Gut microbe-mediated urea nitrogen recycling in plateau pika

To investigate the contribution of gut microbiota to urea nitrogen recycling in plateau pika, we administered an antibiotic cocktail (hereafter Abx) to reduce their gut microbiota populations (Fig 1A). Compared to the control group, pikas in the Abx group exhibited a significant reduction ($p < 0.01$, Student $t$ test) in the $^{13}CO_2$:$^{12}CO_2$ ratio (hereafter $\delta^{13}C$) in breath samples, a finding consistent with microbial involvement in ureolysis (Fig 1B–1D). This Abx treatment also led to lower ($p < 0.05$) plasma urea concentrations but higher ($p < 0.05$) levels in cecal content (Fig 1E and 1F). Concurrently, cecal urease activity and $NH_3$ concentrations were significantly reduced ($p < 0.01$) in the Abx group, while the abundance of the urea transporter UT-B in the cecal epithelium increased ($p < 0.01$) (Fig 1G–1I). Critically, Abx treatment significantly decreased ($p < 0.001$) the incorporation of $^{15}N$ into protein pools in the cecal content, liver, and muscle compared to controls (Fig 1J).

### Seasonal changes in urea nitrogen recycling in plateau pikas

To examine the seasonal dynamics of urea nitrogen recycling in plateau pikas, 8 pikas were captured in each of the summer, autumn, and winter seasons (Fig 2A). Plasma urea concentration was higher ($p < 0.001$) in summer than in autumn

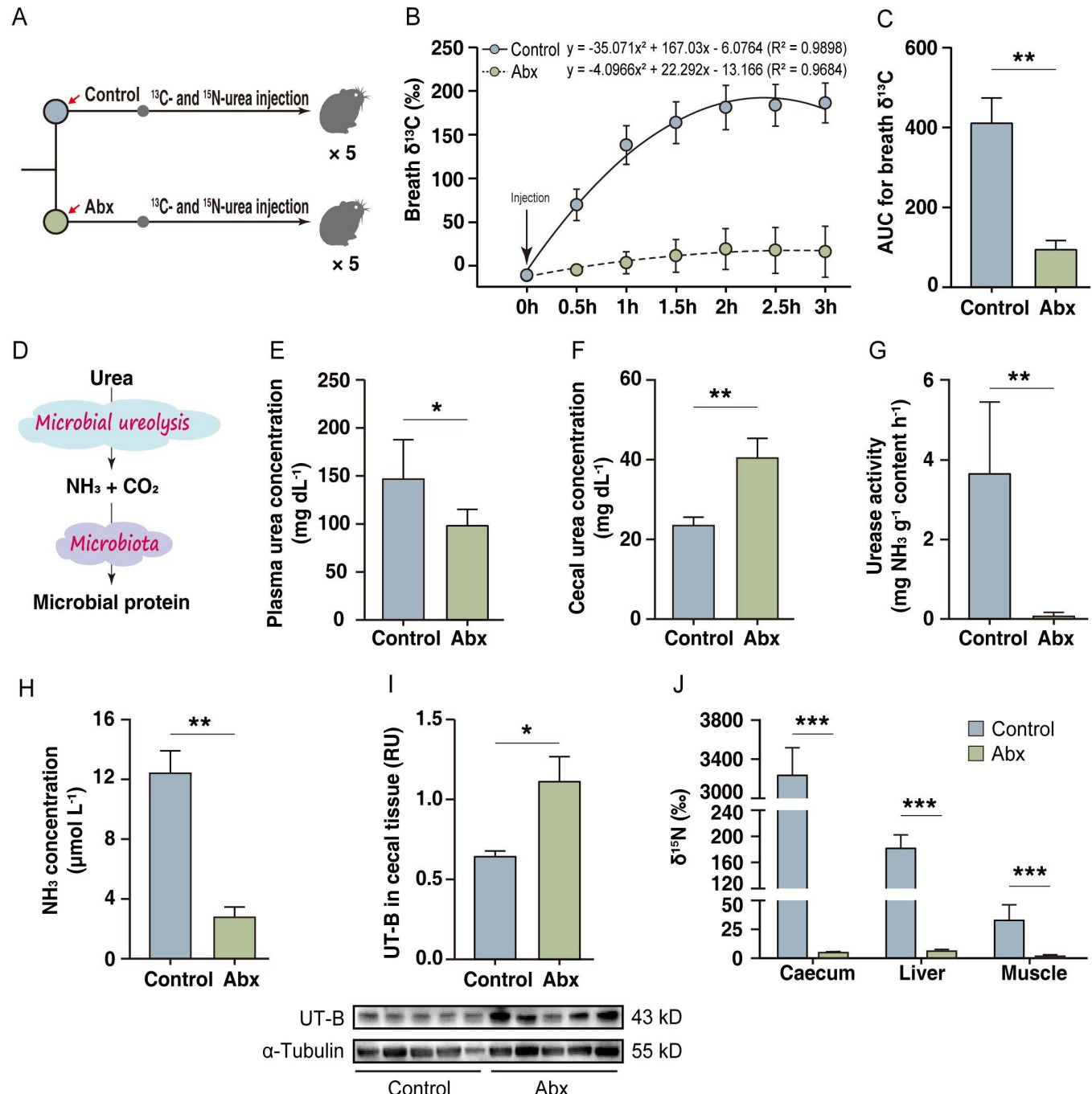

**Fig 1. Gut microbes mediate urea nitrogen salvaging in plateau pikas. (A)** Schematic of the experimental design ($n = 5$ pikas per group). **(B)** Mean breath $\delta^{13}C$ for Abx pikas and control pikas treated with $^{13}C$-urea. The arrows indicate intraperitoneal injection of $^{13}C$-urea. **(C)** Area under the curve (AUC) calculated from the breath $\delta^{13}C$ data in **(B)**. **(D)** Urea hydrolysis in cecal contents. **(E–H)** Effects of microbial clearance on plasma urea concentration **(E)**, cecal urea concentration **(F)**, cecal urease activity **(G)**, cecal $NH_3$ concentration **(H)**, and the abundance of the urea transporter UT-B (western blotting) in the cecal epithelium, as determined by western blotting **(I)**. **(J)** Incorporation of $^{15}N$ from labeled urea into the protein pools of cecal contents, liver, and muscle (quadriceps). The arrows indicate intraperitoneal injection of $^{15}N$-urea. All data represent the mean ± SEM. Statistical significance between the two groups was determined by the two-tailed Student $t$ test. *$p < 0.05$; **$p < 0.01$; ***$p < 0.001$. Abx, antibiotic mixture administered by oral gavage; Control, normal saline administered by oral gavage. The numerical data used to generate the graphs in this figure are available in S1 Data. Raw blot images can be found in S1 Raw Images.

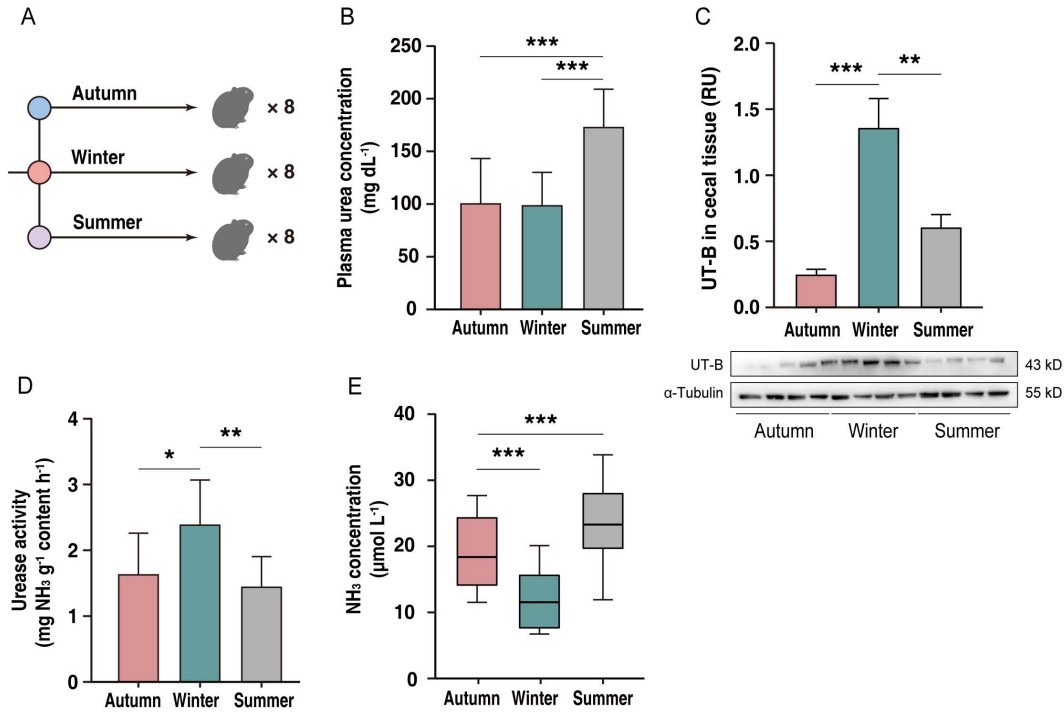

**Fig 2. Seasonal variations in urea nitrogen recycling parameters in plateau pikas. (A)** Schematic of the seasonal sampling design ($n = 8$ pikas per group). **(B–E)** Seasonal effect on plasma urea concentration **(B)**, UT-B abundance in cecal epithelial tissue **(C)**, cecal urease activity **(D)**, and cecal NH$_3$ concentration **(E)**. All data represent the mean ± SEM. Statistical significance was determined by one-way ANOVA with Tukey's post-hoc test. Asterisks denote significant differences among the three groups (*$p < 0.05$; **$p < 0.01$; ***$p < 0.001$). The numerical data used to generate the graphs in this figure are available in S2 Data. Raw blot images can be found in S1 Raw Images.

or winter (Fig 2B), while UT-B abundance in cecal epithelial tissue was greater ($p < 0.01$) in winter than in summer and autumn (Fig 2C). In cecal samples, urease activity was higher ($p < 0.05$), whereas NH$_3$ concentration was lower ($p < 0.05$) in winter than in summer or autumn (Fig 2D and 2E). These seasonal dynamics of urea nitrogen utilization implied that the pikas responded to changes in seasonal exogenous nitrogen by adjusting their endogenous nitrogen utilization.

### Low-protein diet alters the composition and function of the gut microbiota

To examine the effect of dietary protein on urea nitrogen salvaging, 24 pikas were provided one of three diets varying in protein content for 4 weeks (Fig 3A): high protein (HP), containing 18% crude protein (CP); medium protein (MP), 12% CP; and LP, 6% CP (S1 Table). Despite the low protein consumed by the LP group (Fig 3B), the liver weight, ratio of liver weight to body weight, and muscle mass did not differ ($p > 0.05$) among the three groups (Figs 3C, S1A, and S1B); however, the ratio of muscle mass to body weight was higher ($p < 0.05$) in LP than HP pikas (Fig 3D). The concentrations of hepatic carbamyl phosphate (CP), carbamoyl phosphate synthetase 1 (CPS1), and ornithine transcarbamylase (OTC) were higher ($p < 0.01$) in HP than LP pikas (Figs 3E–3G and S1C). LP downregulated ($p < 0.05$) the expression of the genes *Cps1*, *Otc*, *Argininosuccinate synthase 1* (*Ass1*) and *Arginase 1* (*Arg1*) in liver compared with HP, while the expression of *Argininosuccinate lyase* (*Asl*) was not significantly affected (Figs 3H–3K and S1D). The concentrations of plasma and cecal urea and cecal NH$_3$ were lower ($p < 0.05$) (Figs 3M and S1G) whereas UT-B abundance and *Utb* mRNA level in cecal epithelial tissue and cecal urease activity in cecal samples were higher ($p < 0.05$ or $p < 0.001$) in LP than HP pikas (Figs 3L, S1E, and S1F). Hepatic glutaminase activity (S1H Fig) and the acetate concentration in cecal samples (S1I Fig)

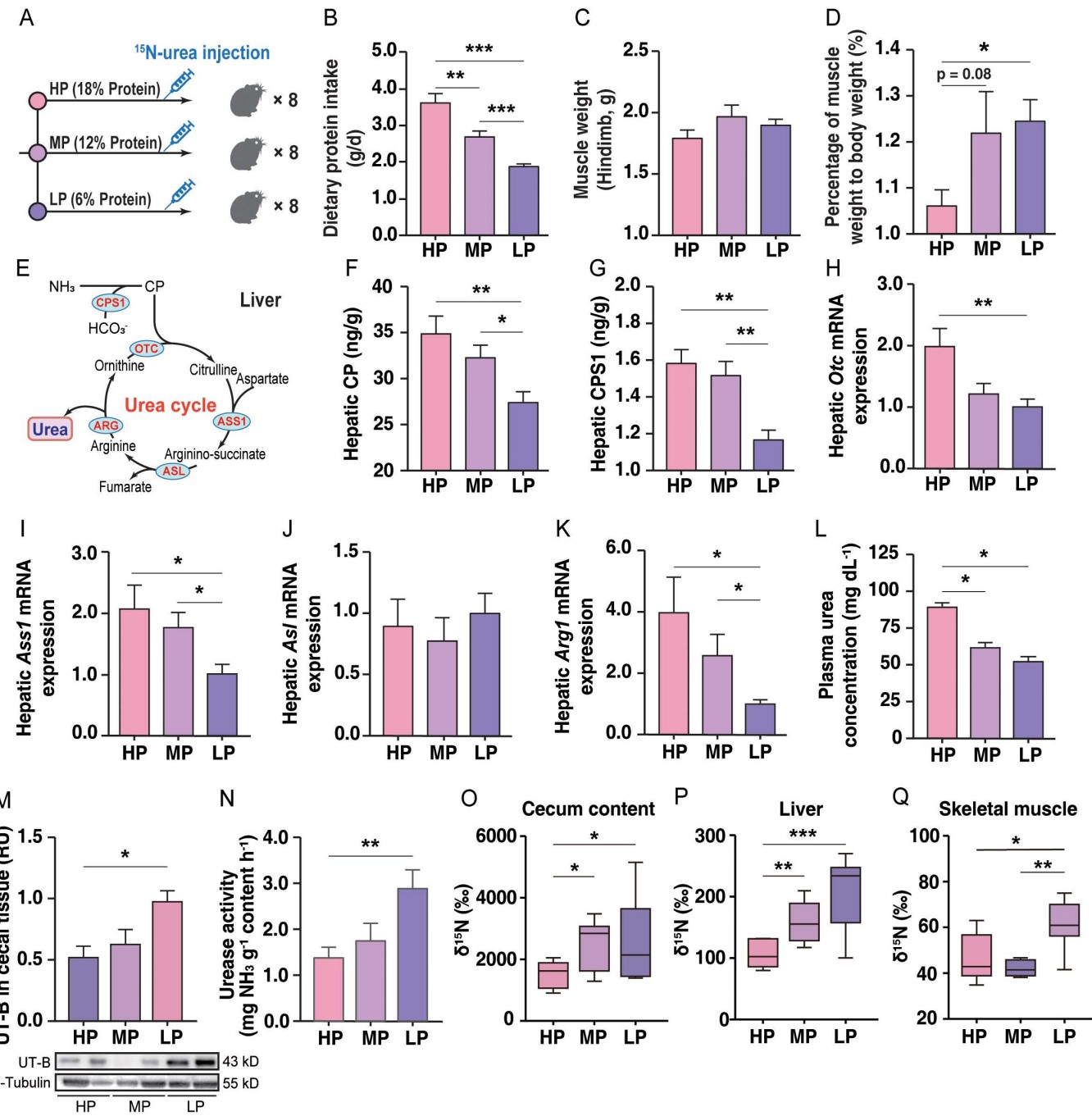

**Fig 3. A low-protein (LP) diet alters metabolic phenotypes and enhances urea nitrogen recycling in plateau pikas. (A)** Schematic of the dietary intervention experiment ($n = 8$ pikas per group). **(B–D)** Effects of dietary protein levels on dietary protein intake **(B)**, muscle weight **(C)**, and the ratio of muscle to body weight **(D)**. **(E)** Hepatic urea synthesis pathway. Effects of hepatic urea synthesis on the concentration of carbamoyl phosphate (CP) **(F)** and carbamoyl phosphate synthetase I (CPS1) **(G)**. **(H–K)** Effect of LP on the expression of *Otc* **(H)**, *Ass1* **(I)**, *Asl* **(J)**, and *Arg1* **(K)** in liver. **(L)** Plasma urea concentration. **(M)** UT-B abundance of cecal epithelial tissue. **(N)** Urease activity in cecal. **(O–Q)** Incorporation of $^{15}$N from labeled urea into the protein pools of cecal contents **(O)**, liver **(P)** and muscle (quadriceps) **(Q)** of pikas fed a high protein diet (HP), medium protein diet (MP), or LP. All data represent the mean ± SEM. Statistical significance was determined by ANOVA with post-hoc Tukey's test. Asterisks denote significant differences among the three groups (*$p < 0.05$; **$p < 0.01$; ***$p < 0.001$). The numerical data used to generate the graphs in this figure are available in S3 Data. Raw blot images can be found in S1 Raw Images.

were higher ($p < 0.01$) in LP than HP pikas. Compared with HP, LP upregulated ($p < 0.05$) the expression of three genes, namely excitatory amino acid transporter 3 (*Eaat3*), L-type amino acid transporter 1 (*Lat1*) and *Lat3* in cecal epithelial tissue (S1J Fig). The results of isotope ratio mass spectrometry (IRMS) revealed that more ($p < 0.05$) $^{15}N$ was incorporated into the cecal content and the protein pool of the liver and muscle in LP pikas than HP pikas (Fig 3O–3Q).

To examine how LP affected the composition and function of ureolytic microbiota, shotgun metagenomic sequencing of genomic DNA in cecal content was performed. The α-diversity, based on species richness, was higher, whereas the Simpson index was lower ($p < 0.01$; Fig 4A) in the LP than HP group. Taxonomic profiling of the metagenomes that was assessed by nonmetric multidimensional scaling (NMDS) based on a Bray–Curtis dissimilarity matrix at the species level (Adonis $p = 0.001$; Fig 4B), demonstrated that the compositions of the cecum bacteria were distinct among the HP, MP, and LP pikas. The metagenomic data were subjected to metabolic pathway analysis with annotation using the Kyoto Encyclopedia of Genes and Genomes (KEGG) database [30]. The two pathways, "arginine biosynthesis" and "other carbon fixation pathways," were enriched in LP pikas compared with HP pikas (false-discovery rate [FDR] < 0.05; Fig 4C and S1 Dataset).

We further examined the functional characteristics of the gut microbiota. Based on a 95% average nucleotide identity threshold [31], 1,382 medium- or high-quality prokaryotic metagenome-assembled genomes (MAGs) were used for subsequent analyses; in each sample, a MAG was considered valid if there was at least 50% genome coverage (Fig 4D and S2 Dataset). Functional annotation of all MAGs by the KEGG database identified 195 MAGs that encoded urease homologs; these belonged to the phyla Bacteroidota (184 MAGs), Bacillota_A (6 MAGs), Desulfobacterota (4 MAGs) and Actinomycetota (1 MAGs) and the genera *CAG-485* (142 MAGs), *Alistipes* (26 MAGs), *CAG-873* (7 MAGs), *JAFLTL01* (6 MAGs) and others (13 MAGs) (Fig 4D and S2 Dataset). Furthermore, an Upset plot analysis revealed that the LP group possessed a larger repertoire of unique urease homologs (40 MAGs) compared to the HP group (15 MAGs), indicating a greater diversity of ureolytic potential in the former (S2A Fig).

We next examined the relative abundances of the MAGs, revealing 11 and 7 MAGs encoding 7 urease homologs (*ureA*, *ureB*, *ureC*, *ureD*, *ureE*, *ureF*, and *ureG*) that were more enriched in LP than in HP (Fig 4F) or MP pikas (S2B Fig). The genera *Alistipes* (*Alistipes* sp. bin1253 and *Alistipes* sp. bin1248) and *CAG-485* (*CAG-485* sp. bin 1085, *CAG-485* sp. bin 1078, *CAG-485* sp. bin 1060, and *CAG-485* sp. 1018) were more enriched (FDR, of < 0.05) in LP than HP and MP pikas (Figs 4E, 4F, S2B, S2C, and S3 Dataset). Moreover, a metagenomic analysis revealed that 150 and 87 MAGs encoding 8 acetate synthetases (EC: 1.2.7.4; EC: 6.3.4.3; EC: 3.5.4.9; EC: 1.5.1.54; EC: 2.3.1.169; EC: 6.2.1.1; EC: 2.7.2.1; EC: 2.3.1.8) of the Wood–Ljungdahl pathway were more enriched (FDR < 0.05) in LP than in HP and MP pikas (S3A–S3C Fig and S4 Dataset), resulting in a higher ($p < 0.05$) acetate concentration in LP than in HP pikas (S3D Fig).

## LPY increases urea nitrogen salvaging induced by LP

To determine whether yak fecal microbiota could enhance urea nitrogen recycling in plateau pikas, we supplemented the pika LP with yak fecal microbiota (now referred to as LPY pikas; Fig 5A). Dietary protein intake (Fig 5B), liver weight, and the ratio of liver weight to body weight did not differ ($p > 0.05$) between the LP and LPY pikas (S4A and S4B Fig); however, muscle mass and the ratio of muscle weight to body weight were lower ($p < 0.05$) in LP than LPY pikas (Fig 5C and 5D). The concentrations of CP, CPS1, and OTC and the expressions of *Cps1*, *Agr1*, *Asl*, and *Ass1* in liver did not differ ($p > 0.05$) between LP and LPY pikas (Fig 5E–5J), with the exception of *Otc* (S4C and S4D Fig). Plasma urea concentration was higher ($p < 0.05$) in LP than in LPY pikas (Fig 5K). Urease activity in cecal samples was lower ($p < 0.05$) in LP than LPY pikas, whereas UT-B abundance and *Utb* mRNA expression in cecal epithelial tissue did not differ ($p > 0.05$) between groups (Figs 5L, 5M, and S4E). Both cecal urea and $NH_3$ concentrations were lower ($p < 0.05$) in LP than LPY pikas (S4F and S4G Fig), but the concentration of short-chain fatty acids (SCFAs) did not differ between groups (S4H Fig). Glutaminase activity in liver did not differ ($p > 0.05$) between LP and HP pikas (S4I Fig). The expression of *Eaat3* and *Lat2* in cecal epithelial tissue was lower ($p < 0.05$) in LP than LPY pikas (S4J Fig). Results from IRMS revealed that less ($p = 0.058$

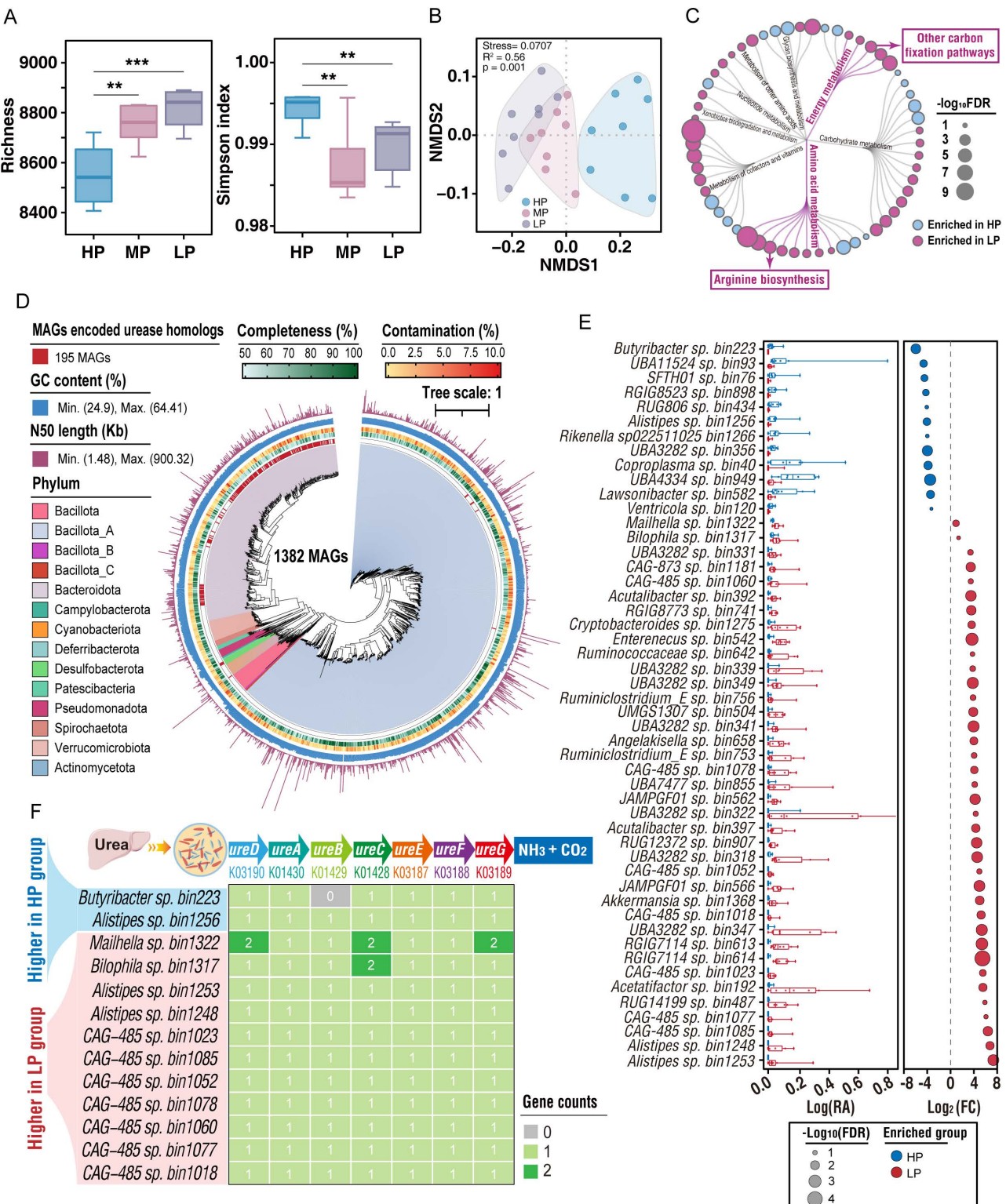

**Fig 4. A low-protein (LP) diet remodels the gut microbiome and enriches for ureolytic functionality. (A)** Alpha diversity of the gut microbiota across the high-protein (HP), medium-protein (MP), and LP diet groups ($n = 8$ per group), measured by the Richness and Simpson index by ANOVA with post-hoc Tukey's test. **(B)** Nonmetric multidimensional scaling (NMDS) plot of Bray–Curtis dissimilarities, illustrating differences in microbial community

structure. Statistical significance was assessed using PERMANOVA (Adonis). **(C)** Enrichment analysis of KEGG pathways comparing the HP and LP groups. Significance was determined using Fisher's exact test with a False Discovery Rate (FDR) correction (FDR < 0.05 indicates a statistically significant difference). **(D)** Phylogenetic tree of 1,382 high-quality metagenome-assembled genomes (MAGs). Rings from the inside out represent: MAGs encoding urease homologs (red); genome completeness (green); genome contamination (yellow); GC content (blue bars); and N50 length (purple bars). Clades are colored by taxonomic classification. **(E)** Comparison of the differential abundance of MAGs from HP pikas vs. LP pikas. The left border represents log-transformed relative abundance (RA, %) of MAGs. The right border represents log-transformed fold-change values of RA. The RA of MAGs between two groups was tested with the nonparametric Wilcoxon test with an FDR-corrected $p$-value (MAGs with FDR < 0.05 are shown). **(F)** The potential of group-enriched MAGs for identifying enzymes encoded by urease homologs (*ureA*, *ureB*, *ureC*, *ureD*, *ureE*, *ureF*, and *ureG*) in HP and LP pikas. The numerical data used to generate the graphs in this figure are available in S4 Data.

or $p < 0.05$) $^{15}$N was incorporated into the cecal content and muscle protein pool of LP than LPY pikas, but there was no difference ($p > 0.05$) in liver tissue (Fig 5N–5P).

The α-diversity, as determined by species richness, was higher ($p < 0.05$) in LP than LPY pikas, whereas the Simpson index did not differ ($p > 0.05$) between groups (Fig 6A). Gut microbiota composition at the species level, as assessed by NMDS, differed (Adonis $p = 0.017$) between LP and LPY pikas (Fig 6B). The pathway "arginine and proline metabolism" was enriched (FDR < 0.05) in LPY pikas compared with LP pikas (Fig 6C and S2 Dataset). Metagenomic analysis revealed that at the phylum level, the relative abundance of Bacteroidota was significantly higher in the LPY group compared to the LP group (S2 Table). This enrichment was driven by significant increases in families such as Prevotellaceae and Rikenellaceae (S3 Table), and correspondingly, the genus *Alistipes* (belong to family Rikenellaceae) was also more abundant (S4 Table). Functionally, this taxonomic shift was substantiated by at the genomic level by the significantly higher abundance of specific MAGs in LPY pikas than in LP pikas, including *Alistipes* sp. (bin1245) and three *CAG-485* sp. bins (*CAG-485* sp. bin1115, *CAG-485* sp. bin1026, and *CAG-485* sp. bin1012) (FDR < 0.1). These MAGs are particularly relevant as they collectively encode seven urease homologs (Fig 6D, 6E, and S3 Dataset). This enhanced functional potential was further confirmed at the gene level; Upset plot analysis showed that the LPY group possessed a larger repertoire of urease homologs (28 unique MAGs) compared to the LP group (21 MAGs) (S4H Fig).

**Responses of gut microbiota-mediated urea nitrogen salvaging to protein restriction in the fecal microbiota transplantation experiment**

To determine the role of gut microbiota in mediating host metabolism, we transplanted fecal microbiota from HP, MP, LP, and LPY pikas into Abx-treated pikas (Fig 7A). Muscle mass and the ratio of muscle mass to body weight did not differ ($p > 0.05$) among pikas fed the three diet regimens that included fecal microbiota transplantation (FMT), i.e., HP microbiota (HP-FMT), MP microbiota (MP-FMT), and LP microbiota (LP-FMT) (S5A and S5B Fig). The concentrations of hepatic CP, CPS1, and OTC were lower ($p < 0.05$) in LP-FMT than in HP-FMT and MP-FMT pikas (Figs 7B, 7D, and S5A). The expression of *Cps1*, *Otc*, and *Ass1* in liver and concentration of plasma urea were lower ($p < 0.05$) in LP-FMT than HP-FMT pikas, but the expression of *Asl* was not significantly affected (Figs 7D–7H, S5C, and S5D). Urease activity in cecal samples and UT-B abundance in cecal epithelial tissue were higher ($p < 0.05$) in LP-FMT than HP-FMT pikas (Figs 7I, 7J, and S5E). The cecal urea and $NH_3$ concentrations were lower ($p < 0.05$) in LP-FMT than HP-FMT pikas (S5F and S5I Fig). In addition, the acetate concentration was higher in LP-FMT than HP-FMT pikas (S5I Fig), although glutaminase activity did not differ ($p > 0.05$) between them (S5H Fig). The expressions of *Eaat3*, *Lat1*, and *Lat2* were higher ($p < 0.05$) in LP-FMT than HP-FMT pikas (S5J Fig). IRMS results revealed that more ($p < 0.05$) $^{15}$N was incorporated into the cecal content and the protein pool in liver and muscle of LP-FMT than HP-FMT pikas (Fig 7K–7M). Metagenomics analysis revealed that the shifts in the recipient pika's cecal microbiota were confirmed by the FMT experiment. The α-diversity, as determined by species richness, was lowest ($p < 0.05$) in LP-FMT pikas, whereas the Simpson index did not differ ($p > 0.05$) among the FMT-HP, FMT-MP, and FMT-LP groups (Figs 7N and S6C). The composition at the species level of cecum microbiota was distinct (Adonis $p = 0.001$) among the FMT-HP, FMT-MP, and FMT-LP groups, as assessed by NMDS (Figs 7O and S6A).

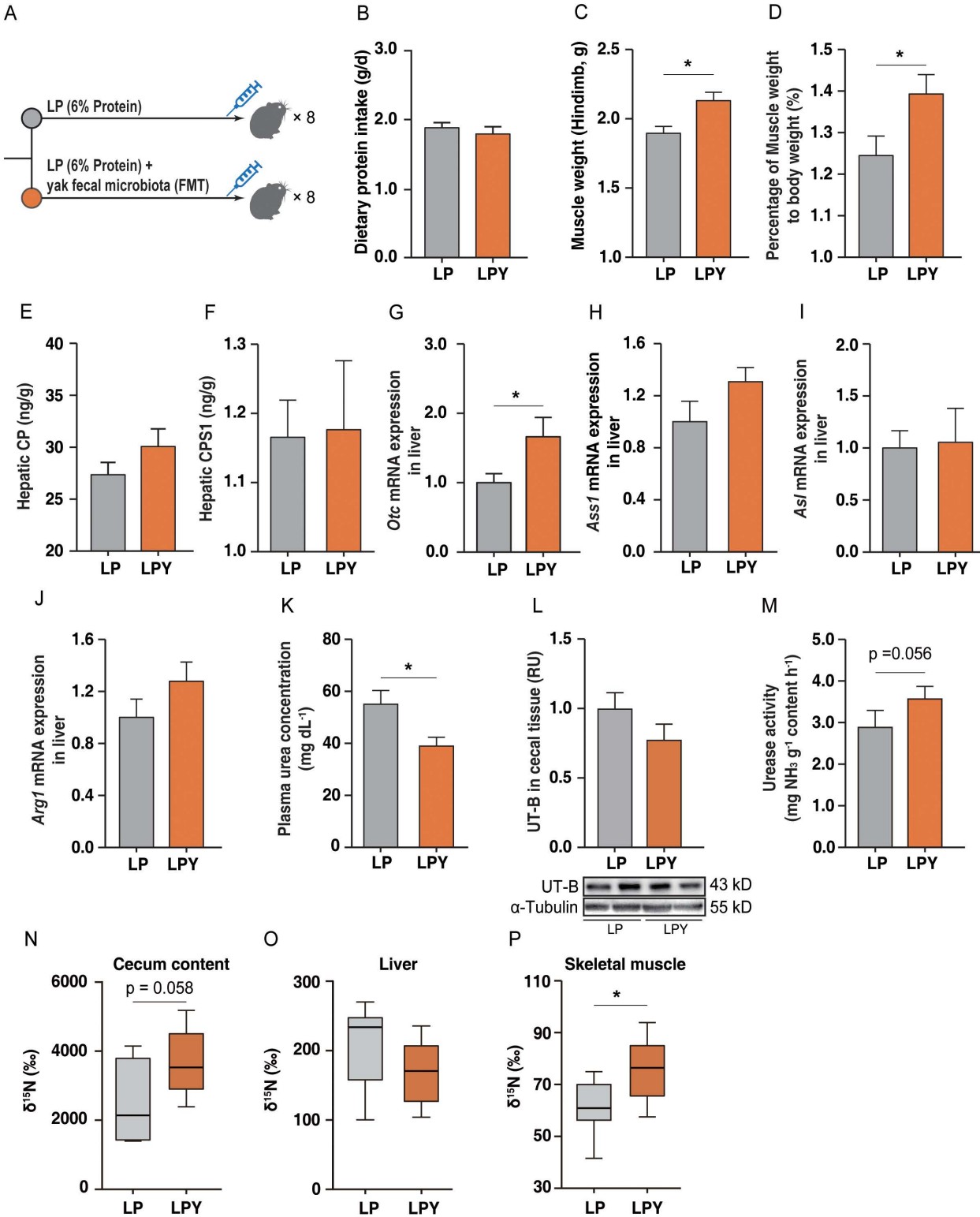

**Fig 5. Fecal microbiota from yaks enhances urea nitrogen recycling in pikas on a low-protein diet. (A)** Schematic of the experimental design, comparing a LP group with a group receiving the same diet supplemented with yak fecal microbiota (LPY; $n = 8$ pikas per group). **(B–D)** General physiological effects on dietary protein intake **(B)**, muscle weight **(C)**, and the ratio of muscle to body weight **(D)**. Effects of hepatic urea synthesis on the

concentration of carbamoyl phosphate (CP) **(E)** and carbamoyl phosphate synthetase I (CPS1) **(F)**. Hepatic urea synthesis of *Otc* **(G)**, *Ass1* **(H)**, *Asl* **(I)** *Arg1* **(J)** gene expression. **(K)** Plasma urea concentration. **(L)** Abundance of the urea transporter UT-B in the cecal epithelium (via western blotting). **(M)** Urease activity in cecal. **(N–P)** Incorporation of $^{15}$N from labeled urea into the protein pools of cecal contents **(N)**, liver **(O)** and muscle (quadriceps) **(P)**. All data represent the mean ± SEM. Statistical significance between the two groups was determined by a two-tailed Student *t* test. Asterisks denote significant differences between the two groups (*$p < 0.05$; **$p < 0.01$; ***$p < 0.001$). The numerical data used to generate the graphs in this figure are available in S5 Data. Raw blot images can be found in S1 Raw Images.

Muscle mass and the ratio of muscle mass to body weight, hepatic CP, CPS1, and OTC abundance, and the expression of *Cps1*, *Otc*, *Agr1*, *Asl*, and *Ass1* did not differ ($p > 0.05$) between LP-FMT and LPY microbiota (LPY-FMT) pikas (Figs 7D–7G and S5A–S5D). Plasma urea concentration, cecal urea, and $NH_3$ concentrations in liver were higher ($p < 0.05$; Figs 7H, S5E, and S5F), whereas urease activity in cecal samples and *Utb* expression in cecal epithelial tissue were lower ($p = 0.07$ or $p < 0.05$; Figs 7I and S5G) in LP-FMT than LPY-FMT pikas. In addition, SCFA (S5I Fig) concentrations and glutaminase activity did not differ ($p > 0.05$) between LP-FMT and HP-FMT pikas (S5H Fig). *Lat2* expression was lower ($p < 0.01$) in LP-FMT than LP Y-FMT pikas (S5J Fig). IRMS results revealed that less $^{15}$N was incorporated into the cecal content and the protein pool of liver and muscle of LP-FMT than LPY-FMT pikas ($p = 0.07$ or $p < 0.05$; Fig 7K and 7M), whereas there was no difference ($p > 0.05$) in liver tissue between groups (Fig 7L). The α-diversity was lower ($p < 0.05$) in FMT-LP than in FMT-LPY pikas, whereas the Simpson index did not differ ($p > 0.05$) between these groups (Figs 7N and S6C). The Bray–Curtis dissimilarity differed ($p = 0.058$) between the LP-FMT and LPY-FMT pikas (Figs 7O and S6B).

## Discussion

Winter-active herbivorous mammals often face the challenge of obtaining sufficient nitrogen with low-protein diets [32]. Nevertheless, the widespread prevalence of herbivory attests to many behavioral and physiological strategies for ensuring nitrogen balance. Nitrogen fixation by gut symbiotic microbes is a key mechanism for increasing nitrogen utilization [33]. On the QTP, seasonal protein content of natural grasses varies widely, and is particularly low in winter, and, therefore, it is challenging for winter-active mammals to maintain protein homeostasis. Our present results demonstrate that urea nitrogen recycled from the liver to the gut and utilized by the microbiota to synthesize protein can benefit these animals. An LP diet in winter altered the composition and function of gut ureolytic bacteria and increased both UT-B abundance and urease activity in cecal samples, with the nitrogen being reincorporated into the pika's protein pool. Our FMT experiment confirmed that LP-induced changes in the microbiota increased the conversion of urea nitrogen into metabolites that could be absorbed by the pikas. Furthermore, FMT and the LPY-induced microbiota enhanced the ureolytic capacity and promoted the entry of ureolytic metabolites into the pika's protein pool, demonstrating a critical mechanism whereby inter-species microbial transfer between sympatric species enhances host protein homeostasis (Fig 8). These findings revealed that microbiota-mediated urea nitrogen salvaging is a critical adaptive strategy, enabling this small mammal to maintain protein homeostasis and thrive in its extreme, high-altitude environment.

Small mammals are sensitive to food availability and quality, especially the protein content of their diets [34,35]. Excess dietary amino acids are preferentially catabolized over carbohydrates and lipids as animals cannot store these amino acids [36]. Urea is hydrolyzed in the gut to $NH_3$ and $CO_2$, and, although toxic, $NH_3$ can be used by microbiota to synthesize proteins [37]. Mammals detoxify $NH_3$, at a considerable energetic cost, through the synthesis of urea via the ornithine cycle [38]. A LP diet that does not satisfy protein requirements reduces both the expressions and activities of urea-cycle enzymes and, consequently, urea levels in liver [39,40]. In the present study, the LP downregulated the relative expression of mRNAs encoding enzymes of the hepatic ornithine cycle, such as *Cps1*, *Ass1*, and *Arg1*, and reductions were also observed in hepatic CP and CPS1 concentrations. Plasma urea concentration can be used to indicate nitrogen utilization and excretion rate [41], with low concentrations suggesting a more efficient use of dietary protein [42,43]. In the present

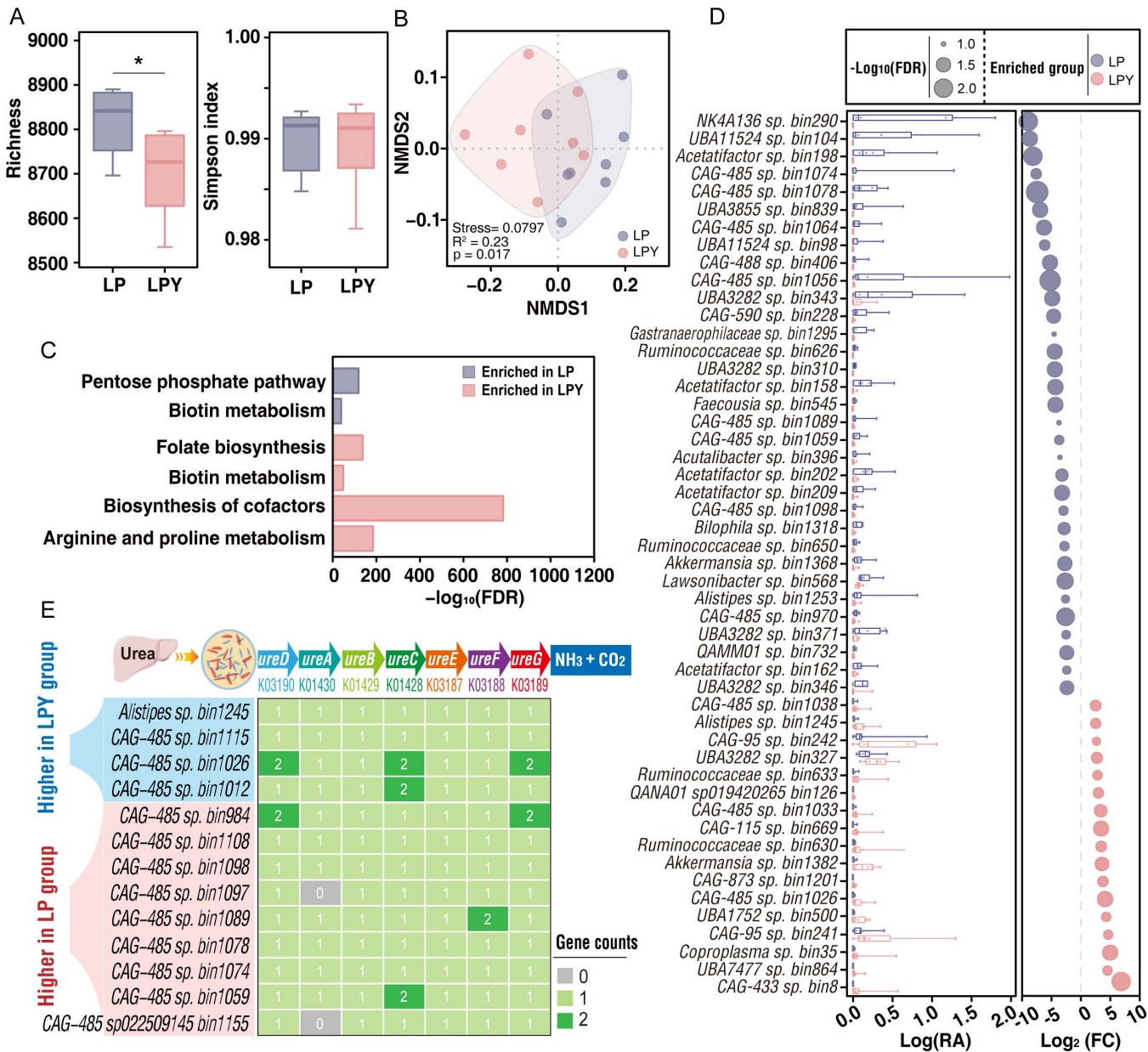

**Fig 6. Fecal microbiota from yaks remodels the pika gut microbiome and enhances ureolytic potential on a low-protein diet.** Metagenomic analysis of cecal microbial communities from pikas on a LP diet vs. those supplemented with yak microbiota (LPY group). **(A)** Alpha diversity of the gut microbiota in the LP and LPY groups ($n = 8$ pikas per group), measured by the Richness and Simpson index. Statistical significance was determined by Student $t$ test ($p < 0.05$). **(B)** Nonmetric multidimensional scaling (NMDS) plot of Bray–Curtis dissimilarities, illustrating the shift in microbial community structure following transplantation. Statistical significance was assessed using PERMANOVA (Adonis). **(C)** Enrichment analysis of KEGG pathways comparing the LP and LPY groups. Significance was determined using Fisher's exact test with an FDR correction (FDR < 0.05 indicates a statistically significant difference). **(D)** Comparison of the differential abundance of MAGs from LP pikas vs. LPY pikas. The left border represents log-transformed relative abundance (RA, %) of MAGs. The right border represents log-transformed fold-change values of RA. The RA of MAGs between two groups was tested with the nonparametric Wilcox tests with an FDR-corrected $p$-value (MAGs with FDR < 0.1 are shown). **(E)** The potential of group-enriched MAGs for identifying enzymes encoded by urease homologs (*ureA*, *ureB*, *ureC*, *ureD*, *ureE*, *ureF*, and *ureG*) in LP and LPY pikas. The numerical data used to generate the graphs in this figure are available in S6 Data.

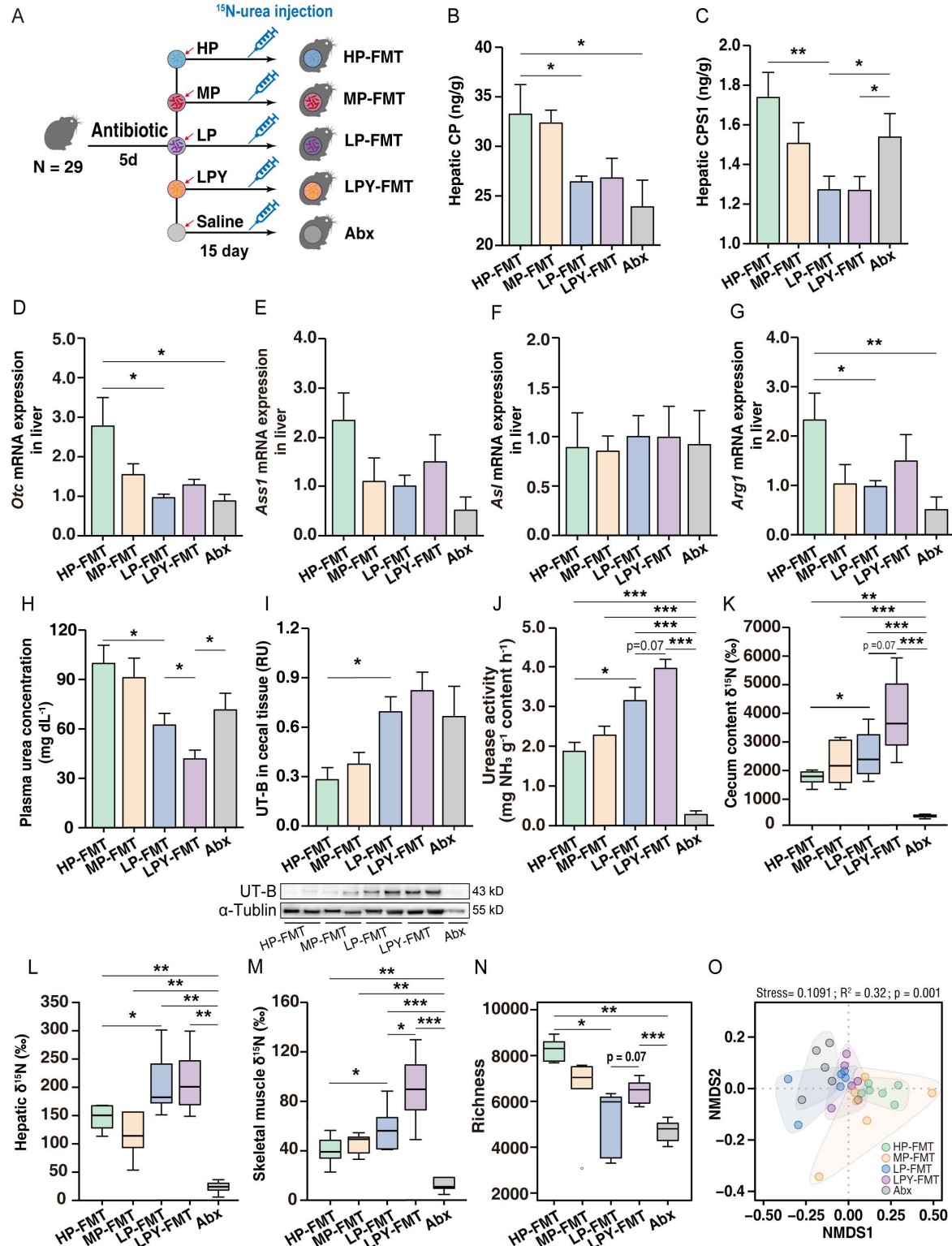

**Fig 7. Fecal microbiota transplantation alters metabolic phenotypes and the diversity and composition of the cecal microbiota. (A)** Schematic overview of the experimental designs. Effects of hepatic urea synthesis on the concentration of carbamoyl phosphate (CP) **(B)** and carbamoyl phosphate synthetase I (CPS1) **(C)**. Hepatic urea synthesis of *Otc* **(D)** *Ass1* **(E)** *Asl* **(F)** *Arg1* **(G)** gene expression. **(H)** Plasma urea concentration. **(I)** Abundance

of the urea transporter UT-B in the cecal epithelium. **(J)** Urease activity in cecal. **(K–M)** Incorporation of $^{15}$N from labeled urea into the protein pools of cecal contents **(K)**, liver **(L)** and muscle (quadriceps) **(M)**. **(N)** Alpha diversity (species richness) of the cecal microbiota. **(O)** Nonmetric multidimensional scaling (NMDS) plot of Bray-Curtis dissimilarities, visualizing community structure differences across all treatment groups. Two experiments are included in this part; one compared metabolic phenotypes and microbial differences induced by different protein diets via FMT experiments (HP-FMT, MP-FMT, LP-FMT and Abx pikas), and the other assessed the effect of supplementation with yak fecal bacteria on metabolic phenotypes and microbes induced by the LP (LP-FMT, LPY-FMT and Abx pikas). The data for LP-FMT and Abx were shared in these two experiments. All data represent the mean ± SEM. $n = 6$ per pikas except Abx pikas ($n = 5$). Statistical significance was determined by one-way ANOVA with Tukey's post-hoc test, where asterisks indicate a significant difference ($p < 0.05$) for the comparison among HP-FMT, MP-FMT, LP-FMT, and Abx groups; for the comparison among LP-FMT, LPY-FMT, and Abx groups, asterisks denote significance at three levels (*$p < 0.05$, **$p < 0.01$, ***$p < 0.001$). The numerical data used to generate the graphs in this figure are available in S7 Data. Raw blot images can be found in S1 Raw Images.

study, although we did not measure urinary urea nitrogen to fully quantify total nitrogen excretion, the plasma urea concentrations of pika in autumn and winter were lower than in summer. This observation is similar to what was observed for the thirteen-lined ground squirrel [10] and free-range ruminants [44], and this may be attributable to the pikas consuming gramineous plants that have a low protein and high fiber content in winter [45]. Likewise, there was a positive correlation between plasma urea concentration and dietary protein intake, with plasma urea concentration being lower in LP than HP pikas. This result implies that protein restriction enhances the utilization of dietary protein or amino-acid in plateau pikas in winter.

Promoting the transfer of urea from the bloodstream into the gut lumen is a potential strategy for improving nitrogen utilization. Functional studies have concluded that the permeability of the gut epithelium to urea and UT-B abundance are key factors in urea regulation [39]. UT-Bs are very responsive to the levels of fermentation products, such as $NH_3$, SCFAs, and $CO_2$ of the gut microbiota, with a negative correlation with $NH_3$ level [46,47]. In the present study, LP upregulated both *Utb* mRNA and UT-B abundance in the cecal epithelium, consistent with higher UT-B abundance in winter than summer. This may be related to the $NH_3$ level in the gut lumen that is reduced with a LP diet during winter. Moreover, microbiota depletion increased UT-B abundance in the cecum, partly owing to an inhibition of urea hydrolysis to yield $NH_3$. Because vertebrates lack endogenous urease [10], the lower $δ^{13}C$ in breath samples after $^{13}C$-urea injection and cecal urease activity in Abx pikas versus control pikas support a role for cecal microbiota in urea hydrolysis. In support, the control pikas incorporated $^{15}$N into liver and muscle protein, whereas Abx pikas did not, thus verifying that urea nitrogen salvaging is microbe-dependent and benefits the host.

Gut microbial communities are critical for the adaptation of wild animals to their habitats, and these communities are linked to the host's energy and protein homeostasis [48]. In mice, an LP diet altered the taxonomic composition of the gut microbiome and also contributed to urea nitrogen recycling [43,49]. The pathway enrichment analysis of the gut microbiota revealed that the microbial genes enriched in response to the experimental diets were related mainly to arginine biosynthesis (including the urea hydrolysis pathway). More of these genes were enriched with the LP than HP diet, indicating that the LP diet establishes a highly selective, nitrogen-limited environment. In such conditions, microbes equipped to hydrolyze urea gain a distinct competitive advantage, allowing them to thrive by utilizing a host-derived nitrogen source. Commensurate with this, the urease activity of the gut microbiota was higher in LP than HP pikas and higher in winter than summer. LP increased urease activity of the gut microbiota in mammals, including mice, European hare, and lambs (*Ovis aries*) [11,43,50]. For mammals, a high level of urease activity implies intensive urea recycling as an adaptation to low nitrogen intake [51]. In addition, the $CO_2$ produced by urea hydrolysis can be utilized via the Wood–Ljungdahl pathway to synthesize acetate as an energy supplement [10]. In the present study, the microbial genes were related mainly to other carbon fixation pathways (including the Wool–Ljungdahl pathway "Converting $CO_2$ into acetate"), and more of these genes were enriched in LP than HP pikas, thus increasing the abundance of acetate in LP pikas and boosting the energy supply. We obtained 192 MAGs encoding urease homologs, accounting for 14% of the total 1,382 MAGs by metagenomic assembly. Of these MAGs, LP increased the abundance of *Alistipes* (*Alistipes* sp. bin1248 and *Alistipes* sp. bin1253) and

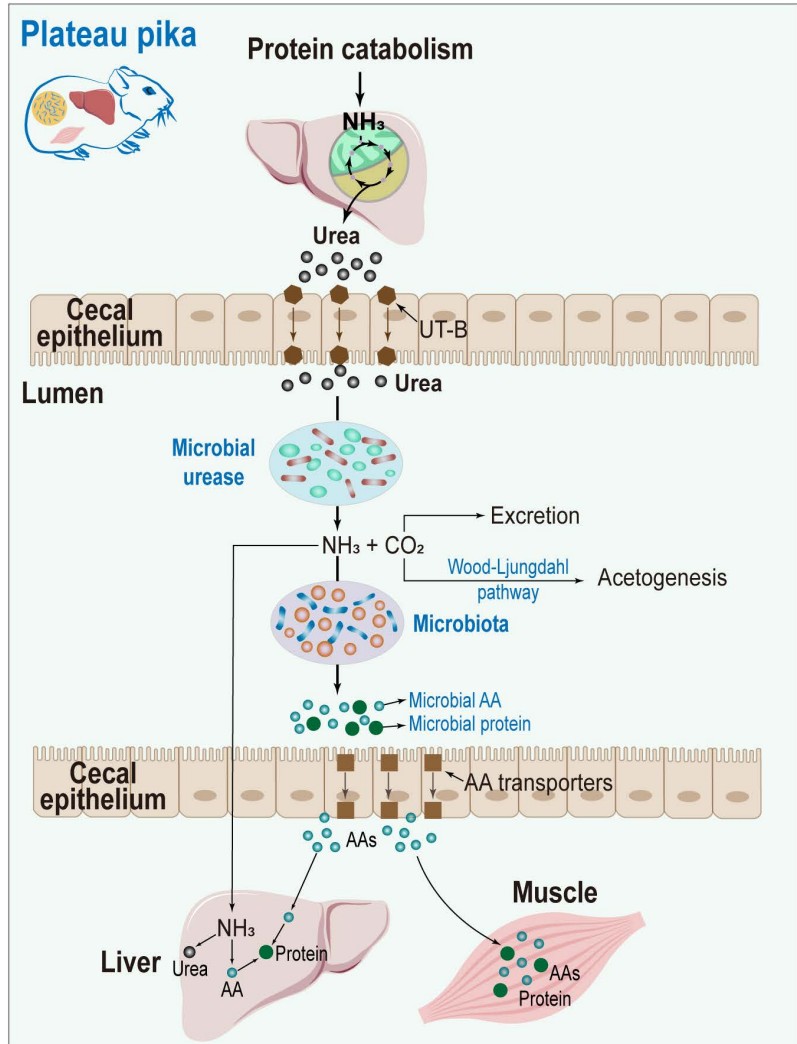

| | LP diet relative to HP diet | LPY diet relative to LP diet |
|---|---|---|
| Hepatic urea synthesis gene abundance | ↓ | → |
| Plasma urea concentration | ↓ | ↓ |
| UT-B abundance | ↑ | → |
| Microbial urease activity | ↑ | ↑ |
| Urease-encoding MAGs | ↑ | ↑ |
| Acetogenesis | ↑ | → |
| Urea N incorporation into microbial protein | ↑ | ↑ |
| AA transporters | ↑ | ↑ |
| Liver glutamine synthetase activity | ↑ | → |
| Urea N incorporation into host protein | ↑ | ↑ |

**Fig 8. Proposed mechanism for urea nitrogen salvaging and relative changes during winter protein restriction.** Urea endogenously produced by the liver is transported by epithelial urea transporters (UT-B) from blood into the cecal lumen where it is hydrolyzed to yield $CO_2$ and $NH_3$ by microbial urease. $CO_2$ is excreted by the host and/or fixed by microbes. $NH_3$ is absorbed by the host and converted into amino acids (AAs) and/or urea in the liver, or it used by microbes to synthesize AAs (green balls) that are incorporated into the microbial proteome or potentially absorbed by the host through amino-acid transporters (brown squares). Ultimately, the AAs are used to synthesize protein (green circles) in host tissues, thus recycling the urea nitrogen. Arrows indicate how processes change (increase, decrease, no change) in HP pikas vs. LP pikas or LP pikas vs. LPY pikas, as revealed by this study.

*CAG-485* (*CAG-485* sp. bin 1085, *CAG-485* sp. bin 1078, *CAG-485* sp. bin 1060, and *CAG-485* sp. 1018) with overrepresentation of urease homologs. *Alistipes* is the main bacterial genus that hydrolyzes urea during hibernation in wild ground squirrels [10,52].

The LP forage on the QTP is an important factor restricting the fecundity of plateau pika [27,53]. Strong evidence indicates that the main benefit of urea nitrogen salvaging is nitrogen incorporation into the protein pool, promoting growth and reproductive capacity [10,14]. The LP pikas incorporated more [15]N into liver and muscle protein than HP pikas, indicating that urea nitrogen salvaging was most beneficial during periods of low protein intake. A significant portion of this salvaged nitrogen was also transferred to the cecal microbiome, leading to demonstrably higher [15]N incorporation into the

microbial protein of LP pikas, a phenomenon that parallels observations in sympatric yaks [54]. The relatively higher $^{15}$N protein in liver of LP pikas may have resulted from the conversion of $NH_3$ to glutamine by glutamine synthetase. A similar nitrogen-conserving mechanism is well-documented in other mammals under protein limitation; for instance, in protein-deficient rats, hepatic $NH_3$ is rerouted from the urea cycle towards glutamine production [55]. Therefore, the strategy for urea nitrogen recycling in the plateau pika appears to be 2-fold, relying on both its gut microbes and its own endogenous mechanisms to ensure nitrogen homeostasis during protein restriction.

Although inadequate consumption of plant protein may lead to amino-acid deficiencies in herbivores, these animals have evolved strategies to mitigate the adverse physiological effects of chronic protein deficiency [56]. Research has demonstrated that food competition between yaks and plateau pikas exacerbates grassland degradation, leading to heightened food scarcity for both species [57]. Paradoxically, areas with higher yak populations often support larger pika populations, suggesting a coexistence mechanism underpinned by coevolutionary adaptations [26,58]. Speakman and colleagues (2021) found that plateau pikas supplement their food intake by eating yak feces, as yak DNA and yak-derived microbes were identified in pika stomach contents in winter, enabling both the body weight and fat content of pika to remain stable across seasons [26]. Evidences indicate that self-coprophagy is extremely common in mammals, where continuous self-inoculation facilitates nitrogen fixation by gut microbes, particularly among herbivores such as hares and northern pikas (*Ochotona hyperborea*) [59,60]. However, interspecific coprophagy, such as plateau pika, is a rare feeding behavior [26]. Our findings reveal this is not merely a nutrient foraging strategy, but an adaptive strategy for functionally augmenting the pika's gut microbiome. Yaks, as large herbivores subsisting on the same LP forage, have gut communities already specialized for efficient nitrogen recycling. Their feces, therefore, serve as a rich source of these preadapted, highly effective microbes. By actively ingesting this source, pikas bypass the slow process of waiting for their own native microbes to adapt. Instead, they directly acquire a superior microbial community tailored to the harsh dietary conditions. The LPY altered the composition and function of gut microbes and enhanced urease activity in the cecal lumen. Specially, the abundance of *Alistipes* species (*Alistipes* sp. bin1245) was more abundant in LPY than LP pikas. *Alistipes* species are key microbes that hydrolyze urea, and *Alistipes* is the dominant genus in the gut lumen of both wild and domestic yaks [61]. Moreover, the greater amount of $^{15}$N that was incorporated into the cecal protein pool in LPY pikas implies that more urea was hydrolyzed and used for microbial protein synthesis, further contributing to microbial growth [62]. In addition, $^{15}$N incorporation into the muscle protein pool in LPY pikas indicated that the supplementation of yak fecal microbiota could improve protein balance by increasing urea nitrogen recycling in winter. We demonstrate a novel facilitative interaction between pikas and yaks on the QTP. This interaction is mediated by interspecific coprophagy, through which pikas obtain key gut microbes from yak feces to improve their ecological adaptability. This microscopic mechanism uncovers the fundamental physiological connection that drives their macroscopic population correlation.

While the collective evidence points to the microbiota's central role in the pika's nitrogen recycling, it is prudent to consider the limitations of our antibiotic model. The Abx employed can exert direct, microbiota-independent effects on host physiology. Evidence indicates that aminoglycosides such as neomycin and streptomycin have been demonstrated to modulate mitochondrial function [63,64], while penicillins may also influence mammalian cellular processes [65]. Consequently, some metabolic alterations observed in the antibiotic-treated group may not be unequivocally attributable solely to microbial depletion. Nevertheless, our primary conclusion regarding urea metabolism remains robust. The marked suppression of ureolysis directly correlated with the elimination of urease-encoding bacteria, providing evidence that this specific phenotype is driven by microbial activity, not a pharmacological artifact. Future investigations are warranted to definitively disentangle these effects using gnotobiotic models and to validate the functional roles of key taxa like *Alistipes* through targeted FMT.

In summary, urea nitrogen salvaging has several benefits for plateau pikas living in the harsh environment of the QTP. First, when dietary nitrogen is insufficient, it enhances host protein synthesis, thereby bolstering the host-protein cache, reducing the energy consumption and foraging risk during winter [66]. Second, the LP, which is typical in winter, limits the

fecundity of plateau pikas, and, thus, the urea nitrogen salvage provides the basis for increasing the amino-acid pool—and hence protein synthesis capacity—until the grass regreening period [53]. Third, the energy demands of pikas increase during winter, and the nitrogen salvage provides energy for pikas to increase thermogenesis and sustain body temperature [25]. Four, the interspecific coprophagy enhancing urea-nitrogen recycling, facilitates resource partitioning between yaks and plateau pikas, reducing dietary competition while promoting microhabitat commensalism through fecal resource tracking, ultimately forming an asymmetric mutualistic relationship.

## Materials and methods

### Animals

All animal procedures were approved by the Animal Care and Use Committee of Shandong University (SYDWLL-2023-067) and implemented in accordance with the national standards of the People's Republic of China (GB/T 35892–2018: Guidelines for the review of animal welfare, GB/T 35823–2018: General requirements for laboratory animal experiments). The capture of wild animals was conducted under a permit (SYXK (Qing) 2022-0001) issued by the Northwest Institute of Plateau Biology, Chinese Academy of Sciences. Given the consistent seasonal fluctuations in forage protein content observed across altitudinal gradients (3,100–5,100 m above sea level) inhabited, we strategically selected the alpine meadow of Laji mountain in Guide County of QTP (3,700 m above sea level; 50 km linear distance from experimental site, <45 min transport duration) for animal capture to minimize translocation stress. Wild plateau pikas were captured at burrow entrances using custom nooses, a technique designed to minimize physical injury and stress. The animals were then transported to the animal laboratory at the Northwest Institute of Plateau Biology, Chinese Academy of Sciences (Xining city, Qinghai Province). Upon arrival, they were immediately housed individually in standard cages (50 cm × 35 cm × 20 cm) and were allowed a 3-week acclimation period to ensure physiological stabilization. This entire period was conducted within a dedicated animal room that was thoroughly disinfected prior to the experiment. The room was maintained under controlled conditions, including a room temperature of 12°C–15°C (reflecting the natural temperature of the experimental environment) and a 12-h light/dark cycle. To minimize cross-contamination, all cages, food, and water were sterilized, and strict hygiene protocols were enforced with restricted personnel access. All animals were monitored daily and remained healthy, showing no signs of infection or distress throughout the experimental period. Their gender and body weight (Body weight in plateau pikas can reflect the animal's age [67]) were assessed, and cages marked accordingly. The pikas were fed standard rabbit pellets (XiaoShu YouTai Biotechnology Co., Beijing, China) and were provided with water ad libitum. The animals were acclimated to the laboratory conditions for 3 weeks prior to experimentation to ensure physiological stabilization.

### Experiment design

**Experiment 1.** To examine the impact of gut microbial clearance on pika urea cycling, 10 acclimated pikas were divided randomly into two groups ($n = 5$ per group). The microbiota-depleted group (Abx-treated) received a daily intragastric gavage of a 400 μl broad-spectrum Abx (50 μg/ml streptomycin, 100 μg/ml neomycin, and 100 U/ml penicillin; Sigma, Germany) for 6 consecutive days. The control group (non-Abx) received an equal volume of sterile saline for the same duration.

The urea isotope tracing protocol followed the method described by Regan and colleagues (2022) [10]. Briefly, each pika received two intraperitoneal injections of a solution containing $^{13}$C-labeled urea (200 mg urea kg$^{-1}$ body mass) and $^{15}$N-labeled urea (200 mg urea kg$^{-1}$ body mass). The two compounds were dissolved together in 1 ml of sterile saline (Sigma-Aldrich, St. Louis, MO, USA) prior to injection. The injections were administered on day 0 (7 days before euthanasia) and again on day 7 (3 h prior to tissue harvesting). On day 7, sampling proceeded around the final urea isotope injection. First, a baseline breath sample was collected immediately before the injection ($t = 0$). Following the injection,

sequential breath samples were collected, beginning at 30 min postinjection and repeated every 30 min thereafter until just prior to euthanasia (at 3 h postinjection) to measure the $\delta^{13}C$ value. Immediately following euthanasia, samples of cecal content, cecal epithelial tissue, liver, and quadriceps muscle were collected. These tissues were snap-frozen in liquid nitrogen and stored at −80°C for subsequent analysis.

**Experiment 2.** To examine the seasonal variations in urea-nitrogen recycling, we captured plateau pikas in each of autumn (October 2021), winter (January 2022), and summer (June 2022). To control for confounding effects of age and body mass, only adult pikas within a narrow body weight range of 130–150 g were selected for the study ($n = 8$ pikas per season). The cecal contents and epithelial tissue were collected immediately after euthanasia and placed immediately in liquid nitrogen, and stored at −80°C until analysis.

**Experiment 3.** To investigate the impact of dietary protein on pika urea nitrogen cycling and cecal microbiota, 24 acclimated pikas were divided randomly into three dietary groups (8 pikas per group, 4 males and 4 females) and each group was offered a diet of different crude protein (see S1 Table): HP (18% CP), MP (12% CP), or LP (6% CP). The experimental diets with graded crude protein levels (18%, 12%, 6% CP) were formulated to simulate the seasonal CP content variation (6%–18% DM basis) observed in alpine meadow vegetation across the QTP [19].

The pikas were maintained on these diets for 4 weeks. At the end of the feeding period, we assessed urea-nitrogen recycling using a $^{15}N$-urea tracing protocol similar to the one described previously. Briefly, pikas received two intraperitoneal injections of $^{15}N$-labeled urea (200 mg urea kg$^{-1}$ body mass), one at the end of week 3 and the final one on the last day of the experiment (Day 28), 3 h before euthanasia. Three hours after the final injection, the animals were euthanized. Samples of blood, cecal content, cecal epithelial tissue, liver, and quadriceps muscle were collected and subsequently analyzed for plasma urea concentration, expression of hepatic urea synthesis genes, urea transporter abundance of cecal epithelial, cecal metabolites (such as SCFAs), and $\delta^{15}N$ in cecal content, liver, and muscle (quadriceps).

**Experiment 4.** To examine the effects of supplementing pika with yak fecal microbiota on microbial metabolites and species of the cecal microbiota, 16 pikas were divided randomly into two groups (8 pikas per group, 4 males, and 4 females). One group was offered the LP diet as in Experiment 3, while the other group was offered the LP diet supplemented with yak fecal microbiota (i.e., LPY pikas). Ten fresh fecal samples of fresh were collected from 10 different adult yaks in areas with pika activity. Approximately 200 mg samples were mixed, then diluted in 2 mL of physiological saline and centrifuged (100 × g for 1 min), and the supernatant was used as yak fecal microbiota for FMT. 400 μL of yak fecal microbiota was transplanted into individually housed pikas by oral gavage daily for 4 weeks. After an additional 4 weeks of acclimation, urea-nitrogen recycling was assessed using the same $^{15}N$-urea tracing protocol as described in Experiment 3. Following the final sample collection, a comprehensive analysis was performed, including measurements of plasma urea concentration, expression of hepatic urea synthesis genes, urea transporter abundance of cecal epithelial, cecal metabolites (such as SCFAs), and $\delta^{15}N$ in cecal content, liver, and muscle (quadriceps).

**Experiment 5.** To examine the influence of gut microbiota on urea-nitrogen cycling, 30 adult pikas were randomly assigned to five groups ($n = 6$ pikas per group; 3 males and 3 females) to serve as transplant recipients. The inoculum for the FMT was prepared under a laminar flow hood as follows: fresh cecal content (200 mg) from donor pikas (HP, MP, LP, and LPY groups) was suspended in 2 mL of sterile 0.9% saline, centrifuged at 500 × g for 1 min, and the supernatant was collected for transplantation. Initially, to deplete the resident gut microbiota, all pikas received a daily oral gavage of a 400 μL antibiotic mixture (50 μg/mL streptomycin, 100 μg/mL neomycin, and 100 U/mL penicillin; Sigma, Germany) for 5 days. Following this, the five groups were transplanted daily for 2 weeks by oral gavage with one of the following: 400 μL of sterile saline (the Abx group), or 400 μL of the prepared supernatant from HP-FMT, MP-FMT, LP-FMT, or LPY-FMT. During the intervention period, one animal in the Abx group was excluded due to health complications deemed unrelated to the experimental treatment, resulting in a final valid sample size of $n = 5$ for this group. All pikas were housed under conditions described in Animals section, and experimental procedures and data collection methods were described in Experiment 3, including measuring food consumption and the effects of treatments on tissues and blood.

## Breath analysis

The protocol for breath analysis was performed as described by Regan and colleagues (2022) [10], with specific procedures as follows. On the day of the breath analysis measurement, each pika was placed in a cylindrical polyethylene metabolic chamber (25 cm long, 10 cm diameter). The chamber was connected via Tygon tubing to a respirometer (Sable Systems FMS-3 gas analyzer, Shenzhen YuanTe Technology Co., Shenzhen, China) to regulate airflow and maintain the $CO_2$ concentration at approximately 1%, ensuring reliable $\delta^{13}C$ measurements.

A baseline breath sample was collected in an aluminum foil air bag (BKMAM, Hunan, China) immediately after the pika was placed in the chamber. Subsequent samples were collected every 30 min. The $^{13}C{:}^{12}C$ ratio of each air sample was determined (termed $\delta^{13}C$, expressed as %) using an elemental analyzer coupled with an isotope ratio mass spectrometer (IRMS; Delta V Advantage, Thermo Fisher Scientific, USA). The resulting $\delta^{13}C$ values, expressed as per mil (‰), were calculated using the standard equation as Regan and colleagues (2022) [10]:

$$\delta^{13}C = \frac{\frac{13c}{12c}\,\text{sample} - \frac{13c}{12c}\,\text{PDB}}{\frac{13c}{12c}\,\text{PDB}}$$

where PDB (Pee Dee Belemnite) served as a standard.

## Analysis of $^{15}N$ incorporation into protein

To determine whether urea-derived nitrogen was incorporated into the host and microbial protein pools, we analyzed the $^{15}N$ enrichment of protein precipitated from cecal content, liver, and quadriceps samples. The methodology was based on the protocol of Regan and colleagues (2022) [10], with specific procedures detailed below.

**Protein extraction from cecal content.** Approximately 50 mg of wet cecal content was homogenized in 350 μl of lysis buffer (10 mM sodium phosphate, pH 7.4) with two homogenization beads. Homogenization was performed using an automatic grinder (Shanghai Jingxin Industrial Development Co., Shanghai, China) with two 45 s pulses at 5.65 m/s, separated by a 30 s interval. An additional 50 μl of lysis buffer was used to rinse the beads, and this volume was combined with the homogenate. The mixture was then centrifuged at 10,000 × g for 10 min at 4°C. The resulting supernatant (~300 μl) was collected, and the protein pellet was obtained after drying in a SpeedVac (Savant, Thermo Fisher Scientific, Waltham, MA, USA) for 40–45 h. Pellets were stored at −80°C until IRMS analysis.

**Protein extraction from tissues.** For liver and quadriceps tissue, a similar procedure was followed. Approximately 50 mg of wet tissue was homogenized in 600 μl of lysis buffer. After centrifugation, the supernatant (~550 μl) was collected, and protein was precipitated by adding an equal volume of ice-cold methanol, followed by a 30 min incubation at −20°C. The mixture was centrifuged again (10,000 × g, 10 min, 4°C) to pellet the protein. The final protein pellet was dried in a SpeedVac or 40–45 h and stored at −80°C for later IRMS analyses.

**Isotope ratio mass spectrometry (IRMS).** For $^{15}N$ analysis, approximately 0.5 mg of each dried protein pellet was weighed into a 3.5 × 5 mm tin cup (Costech, Valencia, CA, USA) and placed into the auto-sampler of a Costech elemental analyzer. The samples were combusted at 980°C in a quartz furnace, and the resulting gases were passed through a reduction furnace (650°C). The purified $N_2$ gas was then separated on a gas chromatography column and introduced into a Delta V stable isotope ratio mass spectrometer (Thermo Finnigan) via a ConFlo III interface (Thermo Finnigan, Bremen, Germany). The ion currents for m/z 28, 29, and 30, followed by m/z 44, 45, and 46, were measured. To ensure accuracy, aliquots of a standard reference material (NIST Standard Material 1547, National Institute of Standards and Technology, US Department of Commerce, Gaithersburg, MD, USA) were analyzed for every 40 experimental samples. The $\delta^{15}N$ values were calculated relative to this standard, with a precision for duplicates averaging 1.47%, where $\delta$ (‰) = $(R_{unknown}/R_{standard} - 1) \times 1000$, and $R$ equals the ratio of the heavy to light isotopes.

## Western blotting

Approximately 0.1 g cecal epithelial tissue was homogenized in RIPA buffer with protease and phosphatase inhibitors, and the homogenate was then centrifuged at 12,000 × g for 10 min at 4°C. Equal amounts of protein in the supernatants were separated using SDS-PAGE on a 10% gel, and the separated proteins in the gel were transferred electrophoretically to a polyvinylidene difluoride membrane. The membrane was incubated three times, each for 10 min, in Tris-buffered saline containing 10% bovine serum albumin and 0.05% (w/v) Tween-20 at room temperature, and was then incubated overnight at 4°C with a primary antibody against one of the following proteins: UTB (UTBc19, diluted 1:3000; provided by Dr. Chongliang Zhong and Prof. Gavin Stewart, University College Dublin) or β-actin (1:1000; Proteintech, 20536-1-AP, Santa Cruz Biotechnology, CA, USA). The secondary antibody was peroxidase-conjugated goat anti-rabbit IgG (Proteintech, SA00001-2, 1:5000). Band intensities were analyzed using Image Lab Software from Bio-Rad, scaled to the intensity measured for β-actin, and expressed as relative units.

## RNA isolation and quantitative real-time PCR

Total RNA was extracted from frozen tissues (liver and cecal epithelial tissue) using TRIzol reagent and the standard phenol-chloroform extraction method. Total RNA was evaluated by 1.5% agarose gel electrophoresis, and the concentration of RNA in solution after extraction of bands was measured using a Nanodrop 2000 spectrophotometer (Thermo Fisher Scientific, Waltham, MA, USA). Total RNA (200 ng) was reverse transcribed to yield cDNA (in 20 μl) using the PrimeScript RT kit (G3337-100, Servicebio, Wuhan, China). Using the cDNA sequence of the plateau pika genome available in GenBank (genome number: GCF_017591425.1), customized primers were designed by Primer 5.0 based on conserved sequences in various genes (see S5 Table). Quantitative real-time PCR was performed using a LightCycler 96 System (Roche Applied Science, Darmstadt, Germany) with SYBR Green real-time PCR Master Mix (AB clonal, RK21203). The amplification efficiency of each target gene and the internal reference gene (*Gapdh*) ranged from 0.97 to 1.07. Relative gene expression was estimated using the $2^{-\Delta\Delta Ct}$ method [68].

## Measurement of urea concentration in plasma and cecal content

Plasma urea concentration was determined using a spectrophotometric assay kit (AKNM002C, Boxbio, Beijing, China). For cecal content, 0.1 g of cecal content was added to 1 ml of extraction buffer (AKNM002C, Boxbio, Beijing, China), centrifuged for 15 min at 12,000 × g and 4°C, and the supernatant was assayed following instructions of the kit.

## Gut microbial urease activity and NH$_3$ concentration

Urease activity and NH$_3$ concentration in cecal content were quantified using a commercial assay kit (AKNM003M, Boxbio, Beijing, China). Following the manufacturer's protocol, approximately 0.1 g of each cecal sample was homogenized in 1.0 ml of extraction buffer while kept in an ice bath. The homogenates were then centrifuged at 12,000 × g for 15 min at 4°C. The supernatant was immediately used for analysis. Urease activity was determined by measuring the absorbance at 630 nm, using a heat-inactivated aliquot of the same sample as a control to correct for nonenzymatic reactions. NH$_3$ concentration was also measured from the same supernatant according to the kit's instructions.

## Liver glutamine synthetase abundance and activity assays

Enzyme activity measurements were made using a commercial spectrophotometric glutamine synthetase activity assay kit (AKAM008M, Boxbio, Beijing, China). Measurement of hepatic CP, CPS1, and OTC content: An enzyme-linked immunosorbent assay kit (Wuhan Fine Biotech Co., Wuhan, China) was used to determine CP, CPS1, and OTC levels in liver samples.

## Metagenomic sequencing and preprocessing

The cecum content of each pika was preserved immediately after euthanization and in a 2 ml cryostorage tube at –80°C. Cecum content underwent bead beating and gel filtration chromatography to obtain total genomic DNA. The QIAamp DNA Stool Mini kit (Qiagen, Hilden, Germany) was used for DNA extraction, following Yu and Morrison (2004) [69]. After assessing DNA purity and concentration via 1.5% agarose gel electrophoresis, DNA libraries were created according to the DNBSEQ protocol [70]. Subsequently, the paired-end raw sequencing reads were filtered and used to construct de novo assemblies [71].

The sequence of pika genomic DNA was obtained (genome number: GCF_017591425.1) using Bowtie2 v2.5.3 [72]. MEGAHIT was used to predict the contigs from each sample, and Prodigal was used to predict the contigs [73,74]. Subsequently, the open reading frames derived from assembled contigs were clustered into a nonredundant dataset by CD-HIT. The abundance profile of genes was calculated and transformed to genes per million, with corrections for variations in gene length and mapped reads per sample.

Metagenomic binning was performed for each sample to obtain MAGs. The initial binning process involved MetaBAT2, Concoct, and Maxbin2 modules that are integrated in MetaWRAP. The assembled contigs were used to build three sets of genomes using MetaBAT2, Concoct, and Maxbin2. These sets were then concatenated and improved by the bin_refinement module of MetaWRAP version 1.3.2. The process of de-replication involved selecting bins and using dRep v3.0.0 with a threshold of 0.95 for Genome Average Nucleotide Identity (commonly known as gANI) (-pa 0.90, -sa 0.95) [75]. The completeness of the bins had to be >50, and the contamination had to be <5 (-comp 50, -con 10), which was determined using CheckM (v.1.0.18) for clustering MAGs. The prediction tool Prodigal (v2.6.3) was used to predict open reading frames [76], which were annotated for function by the Genome Taxonomy Database Toolkit (GTDB-Tk v2.3.2, Database release 214) [77]. The abundance of species per MAG sample was calculated by MetaWRAP and reported in terms of reads per kilobase of transcript per million reads mapped [78].

## Quantification of SCFAs

The concentrations of cecal SCFAs (acetate, propionate, butyrate, isobutyrate, valerate, and isovalerate) were quantified by gas chromatography (7890A column; Agilent Technologies, Santa Clara, CA, USA). Approximately 0.5 g of each cecal content sample was homogenized in 1.5 ml of sterile saline on ice and then centrifuged (10,000 × g, 10 min, 4°C). An aliquot of the supernatant (1 mL) was mixed with 0.2 mL of 25% metaphosphoric acid containing 2-ethylbutyric acid (2 g/L; A506088, Sangon Biotech Co., Shanghai, China) as an internal standard. The mixture was incubated on ice for 30 min and centrifuged again (10,000 × g, 10 min, 4°C). The final supernatant was then analyzed on an Agilent 7890A GC system equipped with a flame-ionization detector and an AT-FFAP capillary column (30 m × 0.25 mm × 0.25 μm). The injector and detector temperatures were 220°C and 230°C, respectively. The oven temperature was programmed as follows: held at 90°C for 1 min, ramped at 10°C/min to 120°C (held for 1 min), and then ramped again at 10°C/min to a final temperature of 150°C (held for 3 min). High-purity nitrogen served as the carrier gas. Individual SCFAs were identified by comparing their retention times with those of a mixed SCFA standard (purity > 99.5%; acetate: No. A801295; propionate: No. P816183; butyrate: No. B802730; isobutyrate: No. I811668; valerate: No. V820439; and isovalerate: No. I811830; all from Macklin, China) and quantified using standard curves generated from serial dilutions of this standard. All results were normalized to the internal standard (S5 Dataset).

## Statistical analyses

The statistical software R version 4.3.3 was used to analyze data. Repeated-measures analysis of variance (ANOVA) was used for dietary protein intake during the 3-week acclimation period. The liver and quadriceps weight, plasma and cecal

urea concentration, cecal urease activity and $NH_3$ concentration, SCFA concentrations, gene and protein expression, and $\delta^{13}C$ and $\delta^{15}N$ measurements were all compared using the Student $t$ test or one-way ANOVA with Tukey's test. Results are presented as the mean ± SEM, and $p < 0.05$ was accepted as significant.

Alpha diversity (Species richness and Simpson index) was compared using one-way ANOVA, while beta diversity (i.e., NMDS) was compared at the species level based on a Bray–Curtis dissimilarity matrix. The relative abundance of each MAG was determined using standard R commands for metagenomics data, and the significance of nonparametric relative abundance profiles was assessed using the Wilcoxon test for two-sample comparisons or the Kruskal–Wallis test for multiple groups. The $p$-values acquired for MAGs were subsequently used to adjust the FDR by the Benjamini–Hochberg algorithm in R, employing the "p.adjust" parameter.

## Supporting information

**S1 Fig. A low-protein (LP) diet alters key metabolic and physiological parameters related to urea nitrogen recycling in plateau pikas.** (A) Liver weight from pikas fed high-protein (HP), medium-protein (MP), or LP diets ($n = 8$ per group). (B) The ratio of liver weight to body weight (BW). (C) Relative mRNA expression of *Cps1* in the liver, as determined by qRT-PCR. (D) Protein abundance of ornithine transcarbamylase (OTC) in the liver, as determined by Elisa. (E) Urea concentration in cecal contents. (F) $NH_3$ concentration in cecal contents. (G) Liver glutamine synthetase (GS) activity in liver. (H) Relative mRNA expression of the urea transporter *Utb* expression in cecal epithelium. (I) Concentration of short chain fatty acids (SCFAs), including acetate, propionate, butyrate, iso-butytate, iso-valerate and valerate in cecal contents. (J) Relative mRNA expression of key amino-acid transporter genes in cecal epithelium. All data are presented as mean ± SEM ($n = 8$ pikas per group). Statistical significance was determined by ANOVA with post-hoc Tukey's test. Asterisks denote significant differences among the three groups (*$p < 0.05$; **$p < 0.01$; ***$p < 0.001$). The numerical data used to generate the graphs in this figure are available in S8 Data.
(TIF)

**S2 Fig. Metagenomic comparison of gut microbiomes between pikas on medium-protein (MP) and low-protein (LP) diets.** (A) Upset plot visualizing the intersections of urease-encoding MAGs among the HP, MP, and LP groups. The bar charts represent the size of each intersection (top) and the total number of MAGs per group (left). (B) Comparison of the differential abundance of MAGs from MP pikas versus LP pikas. The left border represents log-transformed relative abundance (RA, %) of MAGs. The right border represents log-transformed fold-change values of RA. The RA of MAGs between two groups was tested with the nonparametric Wilcoxon test with an FDR-corrected $p$-value (MAGs with FDR < 0.05 are shown). (C) The potential of group-enriched MAGs for identifying enzymes encoded by urease homologs (*ureA*, *ureB*, *ureC*, *ureD*, *ureE*, *ureF*, and *ureG*) in MP and LP pikas. The numerical data used to generate the graphs in this figure are available in S9 Data.
(TIF)

**S3 Fig. $CO_2$ fixation pathways of gut microbiota.** (A) Schematic overview of major $CO_2$ fixation pathways and their key microbial enzymes. (B) Different MAGs that encode enzymes for acetogenesis in $CO_2$ fixation pathways between HP and LP pikas. (C) Different MAGs that encode enzymes of acetogenesis in $CO_2$ fixation pathways between MP and LP pikas. Differences in the relative abundance (RA) of MAGs between two groups were tested with the nonparametric Wilcoxon test with an FDR-corrected $p$-value (MAGs with FDR < 0.05 are shown). The numerical data used to generate the graphs in this figure are available in S10 Data.
(TIF)

**S4 Fig. Yak fecal microbiota supplementation alters metabolic parameters related to urea nitrogen recycling in pikas on a low-protein diet.** (A) Liver weight from pikas in the low-protein (LP) and supplementation of the diet with yak

fecal bacteria (LPY diet). (B) The ratio of liver weight to body weight (BW). (C) Relative mRNA expression of *Cps1* in the liver, as determined by qRT-PCR. (D) Protein abundance of ornithine transcarbamylase (OTC) in the liver, as determined by Elisa. (E) Urea concentration in cecal contents. (F) $NH_3$ concentration in cecal contents. (G) Liver glutamine synthetase (GS) activity in liver. (H) Relative mRNA expression of the urea transporter *Utb* expression in cecal epithelium. (I) Concentration of short chain fatty acids (SCFAs), including acetate, propionate, butyrate, iso-butytate, iso-valerate and valerate in cecal contents. (J) Relative mRNA expression of key amino-acid transporter genes in cecal epithelium (K) Upset plot visualizing the intersections of urease-encoding MAGs between the LP and LPY groups. The bar charts represent the size of each intersection (top) and the total number of MAGs per group (left). All data represent the mean ± SEM ($n = 8$ pikas per group). Statistical significance was determined by the two-tailed Student $t$ test. Asterisks denote significant differences between the two groups (*$p < 0.05$; **$p < 0.01$; ***$p < 0.001$). The numerical data used to generate the graphs in this figure are available in S11 Data.
(TIF)

**S5 Fig. Fecal microbiota transplantation alters metabolic phenotypes of urea nitrogen recycling.** (A) Muscle weight. (B) Ratio of muscle weight to BW. (C) Effects of hepatic urea synthesis on the concentration of OTC in liver. (D) Hepatic urea synthesis of *Cps1* expression. (E) *Utb* expression in cecal epithelium. (F) Cecal urea concentration. (G) $NH_3$ concentration in cecal. (H) Liver glutamine synthetase (GS) activity in liver. (I) Concentration of short-chain fatty acids (SCFAs) in cecal. (J) Expression of genes encoding amino-acid transporters in cecal epithelium. Two experiments are included in this part; one compared metabolic phenotypes and microbial differences induced by different protein diets via FMT experiments (HP-FMT, MP-FMT, LP-FMT, and Abx pikas), and the other assessed the effect of supplementation with yak fecal bacteria on metabolic phenotypes and microbes induced by the LP (LP-FMT, LPY-FMT, and Abx pikas). The data for LP-FMT and Abx were shared in these two experiments. All data represent the mean ± SEM. $N = 6$ pikas except Abx pikas ($n = 5$). Statistical significance was determined by one-way ANOVA with Tukey's post-hoc test, where asterisks indicate a significant difference ($p < 0.05$) for the comparison among HP-FMT, MP-FMT, LP-FMT, and Abx groups; for the comparison among LP-FMT, LPY-FMT, and Abx groups, asterisks denote significance at three levels (*$p < 0.05$, **$p < 0.01$, ***$p < 0.001$). The numerical data used to generate the graphs in this figure are available in S12 Data.
(TIF)

**S6 Fig. Fecal microbiota transplantation alters the diversity and composition of the cecal microbiota.** (A) Nonmetric multidimensional scaling (NMDS) plot of Bray–Curtis dissimilarities, visualizing community structure differences among HP-FMT, MP-FMT, LP-FMT and Abx pikas. (B) NMDS plot based on Bray–Curtis distances representing the differences in the cecal microbial community structure among LP-FMT, LPY-FMT and Abx pikas. (C) Alpha diversity (Simpson index) of bacterial communities. Statistical significance was determined by one-way ANOVA with Tukey's post-hoc test, where asterisks indicate a significant difference ($p < 0.05$) for the comparison among HP-FMT, MP-FMT, LP-FMT, and Abx groups; for the comparison among LP-FMT, LPY-FMT, and Abx groups, asterisks denote significance at three levels (*$p < 0.05$, **$p < 0.01$, ***$p < 0.001$). The numerical data used to generate the graphs in this figure are available in S13 Data.
(TIF)

**S1 Table. Protein content of the various diets.**
(DOCX)

**S2 Table. Average relative abundance (%) of the dominant gut microbial phyla in pikas from the low-protein (LP) and supplementation of the diet with yak fecal bacteria (LPY) groups.**
(DOCX)

**S3 Table. Average relative abundance (%) of the dominant gut microbial family in pikas from the low-protein (LP) and supplementation of the diet with yak fecal bacteria (LPY) groups.**
(DOCX)

**S4 Table. Average relative abundance (%) of the dominant gut microbial genus in pikas from the low-protein (LP) and supplementation of the diet with yak fecal bacteria (LPY) groups.**
(DOCX)

**S5 Table. Primer sequences used for quantitative PCR.**
(DOCX)

**S1 Dataset. Kyoto Encyclopedia of Genes and Genomes (KEGG) pathway enrichment analysis results.** This data contains the detailed results of the KEGG pathway enrichment analysis for two separate comparisons: (Sheet1) High-protein (HP) versus Low-protein (LP) diet groups, and (Sheet2) Low-protein (LP) versus Supplementation of the diet with yak fecal bacteria (LPY) groups. Significance was determined using Fisher's exact test, and pathways with a False Discovery Rate (FDR) corrected $p$-value $< 0.05$ were considered significantly enriched.
(XLSX)

**S2 Dataset. Metagenome-Assembled Genome (MAG) information and urease homolog annotation.** (Sheet 1): MAG assembly and quality statistics. This sheet provides comprehensive assembly and quality information for all 1,382 medium- and high-quality MAGs used in this study. (Sheet 2) MAGs encoding urease homologs. This sheet provides a focused list of all MAGs that were found to encode at least one subunit of the urease enzyme complex (*ureA*, *ureB*, *ureC*, *ureD*, *ureE*, *ureF*, and *ureG*).
(XLSX)

**S3 Dataset. Differentially abundant metagenome-assembled genomes (MAGs) between experimental groups.** This file contains the results of the differential abundance analysis of MAGs for three separate comparisons: High-protein (HP) versus Low-protein (LP) diet groups; Medium-protein (MP) versus Low-protein (LP) diet groups; Low-protein (LP) versus Supplementation of the diet with yak fecal bacteria (LPY) groups. The relative abundance of MAGs between two groups was tested with the nonparametric Wilcoxon test with an FDR-corrected $p$-value.
(XLSX)

**S4 Dataset. Identification of MAGs encoding enzymes of the Wood–Ljungdahl CO₂ Fixation Pathway.** This file details the metagenome-assembled genomes (MAGs) that encode key enzymes for CO₂ fixation via the Wood–Ljungdahl pathway. The dataset includes results from two separate differential abundance analyses: High-protein (HP) versus Low-protein (LP) diet groups; and Medium-protein (MP) versus Low-protein (LP) diet groups.
(XLSX)

**S5 Dataset. Information on the short-chain fatty acid (SCFA) standard compounds and mass spectrometry parameters used for quantification.**
(XLSX)

**S1 Data. Raw date of results in Fig 1.**
(XLSX)

**S2 Data. Raw data of results in Fig 2.**
(XLSX)

**S3 Data. Raw data of results in Fig 3.**
(XLSX)

**S4 Data. Raw data of results in Fig 4.**
(XLSX)

**S5 Data. Raw data of results in Fig 5.**
(XLSX)

**S6 Data. Raw data of results in Fig 6.**
(XLSX)

**S7 Data. Raw data of results in Fig 7.**
(XLSX)

**S8 Data. Raw data of results in S1 Fig.**
(XLSX)

**S9 Data. Raw data of results in S2 Fig.**
(XLSX)

**S10 Data. Raw data of results in S3 Fig.**
(XLSX)

**S11 Data. Raw data of results in S4 Fig.**
(XLSX)

**S12 Data. Raw data of results in S5 Fig.**
(XLSX)

**S13 Data. Raw data of results in S6 Fig.**
(XLSX)

**S1 Raw Images. Western blotting images in Figs 1–3, 5, and 7.**
(PDF)

## Acknowledgments

We appreciate the valuable help from Prof. Jianghui Bian, Prof. Qien Yang, Prof. Jiapeng Qu, and Dr. Guozhen Shang of the Northwest Institute of Plateau Biology, Chinese Academy of Sciences, and from Yourong Lai for field assistance. We also thank the Core Facility and Service Platform, School of Life Sciences, Shandong University, for providing instruments and equipment.

## Author contributions

**Conceptualization:** Fuyu Shi, Yanming Zhang, Dehua Wang.

**Data curation:** Fuyu Shi, Desheng Zou, Liangzhi Zhang, Na Guo, Yuning Ru, Shuai Zheng.

**Formal analysis:** Fuyu Shi, Na Guo, Dehua Wang.

**Funding acquisition:** Fuyu Shi, Dehua Wang.

**Investigation:** Fuyu Shi, Desheng Zou, Liangzhi Zhang, Xianjiang Tang, Shien Ren, Dehua Wang.

**Methodology:** Fuyu Shi, Jiangkun Yu, Shuai Zheng.

**Project administration:** Fuyu Shi, Dehua Wang.

**Resources:** Fuyu Shi, Dehua Wang.

**Software:** Fuyu Shi, Jiangkun Yu, Yuning Ru, Shuai Zheng.

**Supervision:** Yanming Zhang, Dehua Wang.

**Validation:** Fuyu Shi, Dehua Wang.

**Visualization:** Fuyu Shi, Yanming Zhang, Dehua Wang.

**Writing – original draft:** Fuyu Shi, Dehua Wang.

**Writing – review & editing:** Fuyu Shi, Na Guo, Abraham Allan Degen, Dehua Wang.

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
