## [Editor Report · Decision Letter 0]

7 May 2025

Dear Dr Wang,

Thank you for submitting your manuscript entitled "Gut microbiota-mediated urea nitrogen salvaging increases in response to low protein diet during winter in plateau pikas" for consideration as a Research Article by PLOS Biology, and please accept my apologies for the unusual delay in sending you an initial decision. We had wished to discuss your paper with an Academic Editor with relevant expertise, and it took us a bit longer than normal to find someone who was available to provide advice.

Your manuscript has now been evaluated by the PLOS Biology editorial staff as well as by an academic editor with relevant expertise and I am writing to let you know that we would like to send your submission out for external peer review.

Once your full submission is complete, your paper will undergo a series of checks in preparation for peer review. After your manuscript has passed the checks it will be sent out for review. To provide the metadata for your submission, please Login to Editorial Manager (https://www.editorialmanager.com/pbiology) within two working days, i.e. by May 09 2025 11:59PM.

Kind regards,

Luke

Lucas Smith, Ph.D.

Senior Editor

PLOS Biology

lsmith@plos.org

---

## [Decision Letter · Decision Letter 1]

15 Jul 2025

Dear Dehua,

Thank you again for your patience while your manuscript "Gut microbiota-mediated urea nitrogen salvaging increases in response to low protein diet during winter in plateau pikas" was peer-reviewed at PLOS Biology. It has now been evaluated by the PLOS Biology editors, an Academic Editor with relevant expertise, and by several independent reviewers. I would like to apologize again for the protracted review process for your study. As noted, for some reason our reviewer search took a bit longer than normal, and then our third reviewer recently dropped out after some delay. Fortunately, we think that the two reviewers who have provided comments cover the relevant expertise and so we feel comfortable moving forward with inviting a revision based on their assessment of your paper. In lieu of a third reviewer, the Academic Editor has gone through the study in detail and has identified a few additional points that we think should be addressed.

The reviews and comments from the Academic Editor, are appended below. The reviewers agree that the study is potentially interesting and that the experimental design is generally robust. However both reviewers have a number of important suggestions to strengthen the study further. We think that their comments, and the points from the Academic Editor, will need to be thoroughly addressed before we can consider your study for publication at PLOS Biology.

As an additional editorial point, we noticed that there are a few instances of textual overlap between your paper and Regan et al, 2022 (DOI: 10.1126/science.abh2950). After looking into the instances flagged by our system, I see that in many cases the overlap occurs in the 'methods section' of your paper, although I note that there are a couple of sentences in the main text of the manuscript that we think are a bit close for comfort. Below, I am detailing lines where we feel the wording too closely overlaps with the text of Regan et al. In instances where this occurs in the main text, we would require you to rewrite these sentences to put the findings in your own words. For instances where there is overlap in the methods section, we would also require that you edit the text to avoid overlap with Regan et al, to the extent possible, as this could pose a potential copyright issue. Additionally, in cases where you modeled specific methods after those of a different paper (ex the "Isotope Ratio Mass Spectrometry" section), this should be made clearer and the related paper should be cited in the methods section. For example, you could add a note at the start of that section that the 'Isotope ratio mass spectrometry was done following methods developed in (citation)' - before going on to provide a detailed description of what was done.

Below, I detail the instances of textual overlap between your paper and another paper that should be rewritten:

-The sentence in the abstract: "our results reveal a functional role for the gut microbiota in urea nitrogen recycling to maintain protein balance" may be a bit too close to a similar sentence in Regan et al.

-235-237

-291-292

-295-297

-352-355

-398-401

-404-436- the "Isotope ratio mass spectrometry" section

-468-470

Given the extent of revisions needed, we cannot make a decision about publication until we have seen the revised manuscript and your response to the reviewers' comments. Your revised manuscript is likely to be sent for further evaluation by all or a subset of the reviewers.

**IMPORTANT - SUBMITTING YOUR REVISION**

*Re-submission Checklist*

*Published Peer Review*

*PLOS Data Policy*

*Blot and Gel Data Policy*

Sincerely,

Luke

Lucas Smith, Ph.D.

Senior Editor

PLOS Biology

lsmith@plos.org

REVIEWS:

Reviewer #1: The authors investigated how winter protein restriction affects hepatic urea nitrogen recycling in plateau pikas. Their findings demonstrate that gut microbiota-mediated urea nitrogen salvaging is enhanced under low-protein diets. Mechanistically, microbial urease hydrolyzes urea into CO₂ and NH₃, with the absorbed NH₃ subsequently converted into amino acids for potential host uptake via amino acid transporters. While the overall logic is sound and the data generally support the conclusions, several aspects of the manuscript could be improved:

Major Points:

1. Figure 4 identifies differentially abundant microbial taxa between LP and HP groups using metagenomic analysis. Fecal microbiota transplantation (FMT) experiments with key microbial taxa are recommended to validate these metagenomic findings. Similarly, in Figure 6, FMT experiments should be conducted to confirm the role of differentially abundant microbes between LP and LPY groups.

2. The microbial composition of the LPY group is suggested to explicitly describe. Additionally, the diversity of nitrogen-recycling microorganisms in this group should be characterized.

3. How the nitrogen-recycling microorganisms are increased under LP or winter conditions?

4. Additional details regarding experimental conditions, including ambient temperature, environmental sterility levels, and antibiotic treatments administered to the animals, are suggested to describe.

Minor Points:

5. While plasma and cecal urea concentrations were measured, the manuscript does not report urine urea levels. Including these data would provide a more comprehensive assessment of nitrogen excretion and recycling dynamics.

6. To assess seasonal variations in urea nitrogen recycling, pikas were captured across different seasons. However, it is unclear whether pikas in different experimental groups were age- and weight-matched. This information is critical for interpreting group differences.

7. In the Results section, the diet compositions are incorrectly stated as: "HP (high protein), containing 18% crude protein (CP); MP (medium protein), 12% CP; and LP, 18% CP." This appears to be a typo. Based on Figure 3, the LP group should contain 6% CP.

8. Western blot (WB) group labels are missing in Figures 3m, 5I and 7I. These should be added for clarity.

Reviewer #2: This manuscript presents a case study demonstrating enhanced gut microbiota-mediated urea nitrogen salvaging in response to a low-protein diet during winter in a non-hibernating small mammal. The plateau pika (Ochotona curzoniae), a small lagomorph inhabiting the alpine meadows of the Qinghai-Tibetan Plateau, survives the long, cold winter in the face of forage with low protein content. The authors hypothesized that pikas maintain protein homeostasis by enhancing urea nitrogen recycling, particularly during winter when forage protein content is low. Furthermore, winter coprophagy of sympatric yak feces provides plateau pikas with essential nutrients and energy while concurrently altering their gut microbiota composition. Consequently, the authors proposed that this coprophagous behavior increases urea nitrogen utilization, facilitating protein homeostasis maintenance throughout winter. Through integrated application of isotope tracing, metagenomics, and fecal microbiota transplantation, the authors validated their hypotheses. Overall, this study exemplifies a clearly defined scientific question, robust experimental design, and integrated multiple methodologies, resulting in a comprehensive investigation. However, I do have some questions and comments below.

Line 80-83: Is this interspecific coprophagy behavior limited to winter? I suggest you could appropriately introduce a bit more about this behavior. Additional background information on this phenomenon would strengthen the context.

Line 86-90: I recommend refining the hypothesis to sharpen the research focus and avoid a contrived nature. You presented three hypotheses. However, the first two appear to lack sufficient novelty based on current literature for all I know. And the interspecific coprophagy phenomenon represents your most distinctive finding. It may be more attractive for most researchers within the field of zoology, animal behavior, and gut metagenomics, to uncover the role of interspecific corprophagy. You could further develop hypotheses integrating environmental stress (e.g., nitrogen balance), animal behavior (e.g., foraging and interspecific coprophagy), and gut microbiota into a comprehensive theoretical framework.

Line 114: The protein content of LP group is 6%.

Line 201-203: The α-diversity, as determined by species richness, was lowest in LP-FMT pikas, whereas the Simpson index did not differ (p > 0.05) among the FMT-HP, FMT-MP and FMT-LP groups. However, the α-diversity, based on species richness, was higher in the LP than HP group. How do you interpret this phenomenon? Could it be related to the effectiveness of FMT? How do you evaluate the consequence of FMT and its impact on the conclusion?

Line 373-377: How many individuals do these ten fecal samples originate from?

Line 502-508: The methodology in this section could be more detailed. It is necessary to provide the information on standard compounds, and mass spectral target peak in the supporting attachment. Furthermore, given that only six short-chain fatty acids (SCFAs) were analyzed, the use of "metabolomics" in the abstract appears inappropriate. Metabolomic studies typically characterize global metabolic profiles through the quantification of numerous compounds, extending well beyond SCFAs alone.

Figures and tables: There are some minor spelling mistakes inside the figures (e.g., Fig. 4f; Fig. 6; Fig.7, etc). What's the meaning of the group information "CK-FMT" in Fig. 7o? Is it referring to Abx? In addition, consistency in statistical annotations is essential throughout the manuscript. It is not recommended that you used different marking of statistical significance in one paper, especially in a specific figure or subfigure, to ensure clarity and avoid confusion. It is clear enough to use asterisk system (*p < 0.05; **p < 0.01; ***p < 0.001) to display the degree of deviation. Overall, both the figures and the tables should be self-explanatory. All figures (including those in the supplementary information) should have titles and a brief legend explaining their content. Specifically, current supplementary figures lack sufficient legend detail and require expansion for clarity.

Additional points from the Academic Editor:

1. Antibiotics can affect mammal metabolism directly, eg by impacting mitochondrial function and also some have direct effects on animal cells even outside the mitochondria. Therefore, the authors should tone down the claims that changes induced by antibiotics must be due to affects on microbes.

For example they write "The antibiotic treated pikas reduced (p < 0.01, Student's t-test) the 13CO2:12CO2

ratio (hereafter δ13C) in breath samples, indicating that microbes were involved in ureolysis (Figures 1b-1d)."

It would be better first to state that antibiotics can affect mammal cells / physiology directly and then to change the wording to say things like “suggests that microbes were involved”

They should also discuss the possibility that the antibiotics might have direct effects. For example there are many papers on possible direct affects of streptomycin (see https://pmc.ncbi.nlm.nih.gov/articles/PMC12172565/ as an example). Neomycin and penicillin also have effects on mammalian cells.

2. Where the authors discuss annotation it would be better if they stated that they found “homologs” of certain enzymes like urease rather than saying these are ureases since their function has not been characterized.

For example they write "Eg “which encode 7 urease genes” would be better as “which encode 7 urease homologs”

3. From what I can tell the authors are currently releasing the raw sequence data but not the MAGs, which does not meet the standards for data sharing in this field. They need to release / post the MAGs and their annotation to appropriate databases.

---

## [Decision Letter · Decision Letter 2]

12 Sep 2025

Dear Dehua,

Thank you for your patience while we considered your revised manuscript "Gut microbiota-mediated urea nitrogen salvaging increases in response to low protein diet during winter in plateau pikas" for publication as a Research Article at PLOS Biology. This revised version of your manuscript has been evaluated by the PLOS Biology editors, the Academic Editor and the original reviewers.

As you will see in the comments below, the reviewers are satisfied by the revision. Based on their recommendation we are likely to accept this manuscript for publication. However, before we can accept your study we need you to address a number of data and policy related requests, in another revision that we do not think will take very long. These are detailed below.

**IMPORTANT: Please address the following editorial requests

1) TITLE: After discussing the title of your paper within the team, we are wondering if it could be improved by including discussion of the relevance of the increased microbiota mediated nitrogen salvaging. We also think it would be good to indicate that this is in non-hibernating animals. Therefore, if you agree this is supported and appropriate, we suggest you change the title to something like:

"Increased urea nitrogen salvaging by a remodeled gut microbiota helps non-hibernating pikas survive their low-protein winter diet"

2) FINANCIAL DISCLOSURES: Please update your financial disclosures statement, in our editorial manager system, to include the funder websites and a statement indicating whether the sponsors or funders play any role in the study design, data collection and analysis, decision to publish, or preparation of the manuscript.

3) ETHICS STATEMENT:

-- Please include the specific national or international regulations/guidelines to which your animal care and use protocol adhered. Please note that institutional or accreditation organization guidelines (such as AAALAC) do not meet this requirement.

-- I see that you collected wild animals for your study. Please provide the relevant field license number and information on which body granted approval.

4) DATA: Please deposit your raw metagenomic sequencing data on a publicly available repository and update your data availability statement to reference that deposition.

5) DATA: Additionally, you may be aware of the PLOS Data Policy, which requires that all data be made available without restriction: http://journals.plos.org/plosbiology/s/data-availability. For more information, please also see this editorial: http://dx.doi.org/10.1371/journal.pbio.1001797

a. Supplementary files (e.g., excel). Please ensure that all data files are uploaded as 'Supporting Information' and are invariably referred to (in the manuscript, figure legends, and the Description field when uploading your files) using the following format verbatim: S1 Data, S2 Data, etc. Multiple panels of a single or even several figures can be included as multiple sheets in one excel file that is saved using exactly the following convention: S1_Data.xlsx (using an underscore).

b. Deposition in a publicly available repository. Please also provide the accession code or a reviewer link so that we may view your data before publication.

---Regardless of the method selected, please ensure that you provide the individual numerical values that underlie the summary data displayed in the following figure panels as they are essential for readers to assess your analysis and to reproduce it:

Fig 1B-C,E-J; Fig 2B-E; Fig 3B-Q; Fig 4A-B; Fig 5B-P; Fig 6A-D; Fig 8B-O;

FIg S1A-J; Fig S2A,C; Fig S3; Fig S5A-K; FIg S5A-J; Fig S6A-C;

---Please also ensure that figure legends in your manuscript include information on where the underlying data can be found, and ensure your supplemental data file/s has a legend.

----Please ensure that your Data Statement in the submission system accurately describes where your data can be found.

(I see that you do have some data files already accompanying your manuscript, so apologies if some of these data are already provided. Please update these files to make it clear which figure panels they relate to).

6) BLOT AND GEL REPORTING REQUIREMENTS: Please note that we require the original, uncropped and minimally adjusted images supporting all blot and gel results reported in an article's figures or Supporting Information files. We will require these files before a manuscript can be accepted so please prepare and upload them now. Please carefully read our guidelines for how to prepare and upload this data: https://journals.plos.org/plosbiology/s/figures#loc-blot-and-gel-reporting-requirements

7) CODE: Per journal policy, if you have generated any custom code during the course of this investigation, please make it available without restrictions. Please ensure that the code is sufficiently well documented and reusable, and that your Data Statement in the Editorial Manager submission system accurately describes where your code can be found.

We expect to receive your revised manuscript within two weeks.

*Published Peer Review History*

*Press*

Sincerely,

Luke

Lucas Smith, Ph.D.

Senior Editor

lsmith@plos.org

PLOS Biology

Reviewer remarks:

Reviewer #1: The authors have addressed each of the suggestions one by one and refined and organized the manuscript data, making its presentation clearer and content more comprehensive. At present, I have no further revision suggestions for the manuscript.

Reviewer #2: The authors adequately addressed my comments and made substantial revision. I am statisfied with the revision and have no more comments.

---

## [Editor Report · Decision Letter 3]

23 Sep 2025

Dear Dehua,

Thank you for the submission of your revised Research Article "Increased urea nitrogen salvaging by a remodeled gut microbiota helps non-hibernating pikas maintain protein homeostasis during winter" for publication in PLOS Biology and thank you for addressing our last editorial requests in this revision. On behalf of my colleagues and the Academic Editor, Jonathan A. Eisen, I am pleased to say that we can in principle accept your manuscript for publication, provided you address any remaining formatting and reporting issues. These will be detailed in an email you should receive within 2-3 business days from our colleagues in the journal operations team; no action is required from you until then. Please note that we will not be able to formally accept your manuscript and schedule it for publication until you have completed any requested changes.

**IMPORTANT: As discussed over email, I have updated the title and your manuscript file with the most recent versions that you provided me. Please do take a look through your submission to make sure everything looks OK after this change. You will need to edit the paper to remove the highlighting at some point during production.

PRESS

Sincerely, 

Luke

Lucas Smith, Ph.D.

Senior Editor

PLOS Biology

lsmith@plos.org